# To Be Greedy, or Not to Be — That Is the Question for Population Based Training Variants

**Alexander Chebykin**                                           *a.chebykin@cwi.nl*
*Centrum Wiskunde & Informatica*
*Amsterdam, the Netherlands*

**Tanja Alderliesten**                                           *t.alderliesten@lumc.nl*
*Leiden University Medical Center*
*Leiden, the Netherlands*

**Peter A. N. Bosman**                                           *peter.bosman@cwi.nl*
*Centrum Wiskunde & Informatica*
*Amsterdam, the Netherlands*
*Delft University of Technology*
*Delft, the Netherlands*

**Reviewed on OpenReview:** *https://openreview.net/forum?id=3qmnxysNbi*

## Abstract

Achieving excellent results with neural networks requires careful hyperparameter tuning, which can be automated via hyperparameter optimization algorithms such as Population Based Training (PBT). PBT stands out for its capability to efficiently optimize hyperparameter *schedules* in parallel and within the wall-clock time of training a single network. Several PBT variants have been proposed that improve performance in the experimental settings considered in the associated publications. However, the experimental settings and tasks vary across publications, while the best previous PBT variant is not always included in the comparisons, thus making the relative performance of PBT variants unclear. In this work, we empirically evaluate five single-objective PBT variants on a set of image classification and reinforcement learning tasks with different setups (such as increasingly large search spaces). We find that the Bayesian Optimization (BO) variants of PBT tend to behave greedier than the non-BO ones, which is beneficial when aggressively pursuing short-term gains improves long-term performance and harmful otherwise. This is a previously overlooked caveat to the reported improvements of the BO PBT variants. Examining their theoretical properties, we find that the returns of BO PBT variants are guaranteed to asymptotically approach the returns of the *greedy* hyperparameter schedule (rather than the optimal one, as claimed in prior work). Together with our empirical results, this leads us to conclude that there is currently no single best PBT variant capable of outperforming others both when pursuing short-term gains is helpful in the long term, and when it is harmful.

## 1 Introduction

Neural network training requires setting values of many hyperparameters, a process that is both crucial for good performance and time-consuming. Hyperparameter Optimization (HPO) algorithms aim to automate Hyperparameter (HP) tuning (Bergstra et al., 2011; Feurer & Hutter, 2019).

The Population Based Training (PBT) algorithm (Jaderberg et al., 2017) and its variants (Parker-Holder et al., 2020; 2021; Wan et al., 2022; Dalibard & Jaderberg, 2021) stand out for several reasons. Firstly, they are designed to efficiently optimize a schedule of HPs (rather than one set of fixed values), which improves performance (Loshchilov & Hutter, 2017; Tan & Le, 2021). Unlike other dynamic HPO algorithms (Baydin

et al., 2018; Badia et al., 2020; Li et al., 2022), PBT is general, i.e., not restricted to a specific HP such as learning rate, or to a specific setting such as Reinforcement Learning (RL). Secondly, PBT runs within the wall-clock time of a single training of a network, which is desirable in practice. Thirdly, PBT is convenient to scale: adding parallel workers improves results without strongly influencing the wall-clock time of the algorithm. Finally, PBT can directly optimize non-differentiable objectives of interest, such as accuracy of a classifier, or score of an RL agent.

PBT was built upon in Population Based Bandits (PB2, Parker-Holder et al. (2020)) where Bayesian Optimization (BO) was introduced for sampling new HP values, PB2-Mix (Parker-Holder et al., 2021) where the BO setup of PB2 was extended to include discrete HPs, Bayesian Generational PBT (BG-PBT, Wan et al. (2022)) where a more advanced BO approach was used, and Faster Improvement Rate PBT (FIRE-PBT, Dalibard & Jaderberg (2021)) where the greedy nature of PBT was addressed. In each publication, the proposed variant outperformed previous variants included in the experiments, which unfortunately did not always include the best previous variant. Additionally, each subsequent publication performed evaluations on different tasks and in different setups (e.g., different implementations of RL tasks, different computation budgets).

These issues make it difficult to say if one of the PBT variants is substantially better than all others, and should therefore be preferred in practice. The question of whether there is one superior algorithm is the key question we aim to answer in this work. Towards that goal, we conduct an impartial empirical evaluation of the PBT variants in image classification and RL settings, using the FashionMNIST (Xiao et al., 2017), CIFAR-10/100 (Krizhevsky & Hinton, 2009), and TinyImageNet (Stanford CS231N, 2017) datasets for the former, and a subset of Brax tasks (Freeman et al., 2021) for the latter. Furthermore, we look into the theoretical foundations of the BO variants of PBT to better understand the proved guarantees and how they correspond to the behaviour of the algorithms in practice.

Although there exist extensions of PBT for multi-objective optimization (Dushatskiy et al., 2023) and architecture search (Franke et al., 2020; Wan et al., 2022; Chebykin et al., 2023), the focus of this article is on single-objective optimization of HPs only. Furthermore, we do not aim to compare PBT variants (which we also refer to as PBTs) to other HPO algorithms.

The main contributions of our paper are threefold:

- We investigate whether the more recently introduced PBT variants are always superior to the earlier ones on a set of image classification and RL tasks.

- We study how the performance and runtime of the PBT variants change when the task setup is varied.

- Theoretical and mechanistic causes for the observed differences in performance are explored.

## 2 Dynamic HPO

### 2.1 Problem statement

We formalize the problem setting by building upon the notation of Jaderberg et al. (2017). In the single-objective setting, the goal is to maximize the final performance of a neural network by optimizing a schedule of HPs. Let us denote the HP search space as $H$, the HPs used during training between times $t$ and $t+1$ as $\boldsymbol{h}_t \in H$, the maximum time as $T+1$, the weight space as $W$, the weights as $\theta_t \in W$, the objective function as $Q : W \to \mathbb{R}$, and the training procedure as $\texttt{train} : W \times H^T \to W$. The dynamic HPO optimization problem can be written as:

$$\{\boldsymbol{h}_t^*\}_{t=1}^T = \underset{\{\boldsymbol{h}_t\}_{t=1}^T}{\arg\max} \, Q(\texttt{train}(\theta_1|\{\boldsymbol{h}_t\}_{t=1}^T)). \tag{1}$$

Note that while Eq. 1 formalizes *dynamic* HPO in that $\boldsymbol{h}$ varies over time, the optimization is written as *static* or non-sequential (arg max is applied to the entire schedule) to correspond to the informal problem

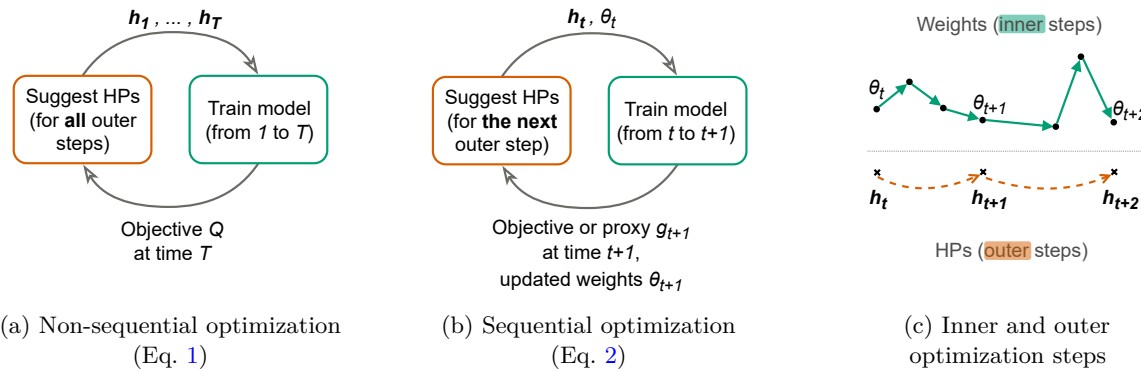

Figure 1: Illustration of concepts relevant to the problem statement (see Section 2.1).

statement of maximizing the *final* performance. Drawing the distinction between optimizing a schedule statically (non-sequentially) or dynamically (sequentially) is important for explaining our theoretical results in Section 4. In Figure 1, panels (a) and (b) illustrate the non-sequential and sequential optimization loops.

For *sequential* optimization, consider $\texttt{step} : W \times H \to W$, a logical (outer) step of a PBT algorithm that corresponds to multiple inner steps (in which the weights are updated via, e.g., gradient descent). The hyperparameters are updated after each outer step. We can expand $\texttt{train}(\theta_1 | \{\boldsymbol{h}_t\}_{t=1}^{T}) = \texttt{step}(\texttt{step}(\dots \texttt{step}(\theta_1 | \boldsymbol{h}_1) \dots | \boldsymbol{h}_{T-1}) | \boldsymbol{h}_T)$. If we denote the objective or its proxy after $t$ $\texttt{steps}$ as $g_t : H^t \to \mathbb{R}$, dynamic sequential HPO can be written as a sequence of problems:

$$\{\tilde{\boldsymbol{h}}_t\}_{t=1}^{T} = \{\arg\max_{\boldsymbol{h}_1} g_1(\boldsymbol{h}_1), \arg\max_{\boldsymbol{h}_2} g_2(\boldsymbol{h}_2 | \tilde{\boldsymbol{h}}_1), \dots, \arg\max_{\boldsymbol{h}_T} g_T(\boldsymbol{h}_T | \tilde{\boldsymbol{h}}, \dots \tilde{\boldsymbol{h}}_{T-1})\}. \tag{2}$$

The sequential nature of optimization entails that previous choices cannot be altered. Consequently, directly optimizing $Q$ at each step would make the optimization greedy, while optimizing an alternative $g_t$ could lead to better long-term outcomes (as showcased by FIRE-PBT, see Section 3.1).

Let us further denote by $\text{step}_{\text{inner}}$ the number of inner optimization steps taken within each outer $\texttt{step}$. Since $\texttt{step}$ is applied $T$ times, the total number of inner optimization steps is $\text{total}_{\text{inner}} = T \cdot \text{step}_{\text{inner}}$. Figure 1c shows an example where each outer $\texttt{step}$ corresponds to $\text{step}_{\text{inner}} = 3$ inner steps. For the visualized two outer $\texttt{steps}$, the total number of inner optimization steps $\text{total}_{\text{inner}}$ is equal to 6.

## 2.2 Dynamic HPO algorithms

In this section, we provide an overview of dynamic HPO algorithms that are not PBT variants. While there are many HPO algorithms that tune static values of HPs (Bergstra et al., 2011; Li et al., 2018; Falkner et al., 2018; Awad et al., 2021; Tan et al., 2024), few can efficiently tune schedules of HPs (although any HPO algorithm can learn an HP schedule by introducing separate variables, one for each HP at each time step, this is unlikely to be efficient). One approach to dynamic HPO is to adjust the HPs via hypergradient descent (Baydin et al., 2018; Li et al., 2022). While using hypergradients can be efficient, it is also restrictive of HPs that can be tuned this way: e.g., only learning rate (in (Baydin et al., 2018)) or HPs that are both continuous and differentiable (in (Zahavy et al., 2020; Li et al., 2022)). Additionally, hypergradient algorithms introduce HPs of their own that require tuning (Paul et al., 2019; Jin et al., 2021).

Another research direction focuses on dynamic HPO specifically for RL, which can be performed via bandit optimization (Badia et al., 2020), evolutionary methods (Tang & Choromanski, 2020), hypergradients (Zahavy et al., 2020), or by estimating HP quality via weighted importance sampling (Paul et al., 2019).

Biedenkapp et al. (2020) use RL for dynamic optimization in the general setting of dynamic algorithm configuration. While promising, the approach relies on RL that requires a large number of roll-outs to learn,

thus making it too inefficient for dynamic HPO on time-consuming tasks such as neural network training (e.g., it required 40,000 training runs in the practical setting of tuning CMA-ES (Adriaensen et al., 2022)).

Unlike the approaches discussed above, the PBT variants are general algorithms that were empirically shown to be capable of tuning arbitrary HPs on challenging tasks. We further discuss them in Section 3.

# 3 PBT variants

## 3.1 Exploration via perturbation ("Perturbation PBTs")

**PBT** Jaderberg et al. (2017) introduced the first PBT algorithm for dynamic HPO. In PBT, the training of $N$ models is interleaved with HPO. A solution $p_i$ at (logical) time $t$ is a tuple of model weights and HPs: $(\theta_t^i, \boldsymbol{h}_t^i)$. A population $P = \{p_i\}_{i=1}^N$ encompasses all solutions. At each time $t = 1 \ldots T$, all models are trained in parallel[1] for step$_\text{inner}$ inner steps corresponding to one outer step: $\theta_{t+1}^i = \texttt{step}(\theta_t^i | \boldsymbol{h}_t^i)$, and evaluated: $q_{t+1}^i = Q(\theta_{t+1}^i)$.

After evaluation has been performed at time $t$, the `exploit` procedure is applied. Jaderberg et al. (2017) experimented with several `exploit` procedures and found truncation selection to perform best, which was then used in all the PBT variants we describe. In truncation selection, each of the $\lambda\%$ worst-performing models is replaced with a copy of one of the $\lambda\%$ best-performing ones (chosen randomly).

Finally, the `explore` procedure is applied to the copies created by the `exploit` procedure. In PBT, real-valued HPs are explored via perturbation: each HP is multiplied by a randomly chosen factor from a predefined set (e.g., $\{0.5, 2.0\}$); categorical HPs are randomly resampled. After the `exploit` step, the interleaved training and HPO continues with the updated population (as described above).

**FIRE-PBT** Dalibard & Jaderberg (2021) observed that PBT can behave too greedily, potentially harming its long-term performance. This is addressed by splitting the population into hierarchical subpopulations $P_{1..J}$ and a subpopulation of "evaluators". All subpopulations $P_{1..J}$ execute the standard PBT algorithm (in parallel) but optimize different objective functions. $P_1$ directly optimizes the target objective $Q$. Each subsequent subpopulation $P_j$ optimizes an Improvement Rate (IR) objective defined relative to $P_{j-1}$. Omitting details, for $(^j\theta_t^i, {}^j\boldsymbol{h}_t^i) \in P_j$, IR measures how fast $Q$ improves when the weights $^j\theta_t^i$ are trained with the currently best HPs from $P_{j-1}$: $^{j-1}\boldsymbol{h}_t^*$. The "evaluators" enable the computation of IR by performing this training. Additionally, the weights trained by an evaluator can replace the weights associated with $^{j-1}\boldsymbol{h}_t^*$, thus allowing the weights from $P_j$ to propagate to $P_{j-1}$. Using IR instead of $Q$ in the subpopulations $P_{2..J}$ makes optimization less greedy, see Dalibard & Jaderberg (2021) for further details.

## 3.2 Exploration via Bayesian optimization ("Bayesian PBTs")

**PB2** Parker-Holder et al. (2020) replace the random perturbation `explore` procedure of PBT with a BO approach based on Time-Varying Gaussian Process Bandits (TV-GPB) (Bogunovic et al., 2016). At each time $t$, changes in performance relative to the previous step are computed: $y_t^i = q_t^i - q_{t-1}^i$, and collected in a dataset: $D_t = D_{t-1} \cup \left\{ \left( y_t^i, t, \boldsymbol{h}_t^i \right) \right\}_{i=1}^N$. Afterwards, a Gaussian Process (GP) model is fit to the dataset and new HPs are selected by optimizing an acquisition function that extends TV-GPB to a parallel setting. The theoretical guarantees for PB2 (and other Bayesian PBTs) are based on the TV-GPB formalism that is further discussed in Section 4.

**PB2-Mix** The BO model in PB2 is defined only for real-valued hyperparameters, while the categorical HPs are randomly resampled. To include categorical variables into the BO setup of PB2, Parker-Holder et al. (2021) introduce the `TV.EXP3.M` multi-armed bandit algorithm for the time-varying setting. `TV.EXP3.M` is used to sample the categorical variables, after which a time-varying GP model relying on a joint categorical-continuous kernel is learned and used to sample the continuous variables.

---

[1] Note that we describe a synchronous version of PBT.

**BG-PBT**  BG-PBT (Wan et al., 2022) builds upon the previous Bayesian PBT variants by explicitly considering ordinal variables in its extended version of the CASMOPOLITAN algorithm (Wan et al., 2021). This algorithm was chosen for its suitability to high-dimensional and mixed-input problems thanks to adaptations such as trust regions. BG-PBT places focus on enabling neural architecture search, which is out of scope for our article, we therefore omit the related adaptations from our analysis and experiments (which therefore apply to a variant of BG-PBT without architecture search).

## 4  Analysis

### 4.1  The returns of the Bayesian PBTs asymptotically approach the returns of the greediest schedule

Parker-Holder et al. (2020) formalize the problem of dynamic HPO as optimizing a time-varying objective function $F_t(\boldsymbol{h}_t)$. In our notation, it is equivalent to $Q(\mathtt{train}(\theta_1|\boldsymbol{h}_t, \{\tilde{\boldsymbol{h}}_i\}_{i=1}^{t-1}))$: evaluating the weights trained with a schedule where the HPs before time $t$ are fixed, while the HPs at $t$ can be varied. The stated goal is to optimize the final performance $F_T(\boldsymbol{h}_T)$ with respect to a hyperparameter schedule $\{\boldsymbol{h}_t\}_{t=1}^T$. The optimization is sequential: the best choice at each step is defined as $\tilde{\boldsymbol{h}}_t = \arg\max_{\boldsymbol{h}_t} f_t(\boldsymbol{h}_t)$, where $f_t$ is the improvement in $F_t$ after one outer $\mathtt{step}$ of an algorithm: $f_t(\boldsymbol{h}_t) = F_t(\boldsymbol{h}_t) - F_{t-1}(\boldsymbol{h}_{t-1})$. This corresponds to the dynamic sequential HPO setup in Eq. 2 with $g_t = f_t$. The regret is defined as $\tilde{r}_t(\boldsymbol{h}_t) = f_t(\tilde{\boldsymbol{h}}_t) - f_t(\boldsymbol{h}_t)$. Let us reproduce Lemma 1 of Parker-Holder et al. (2020):

**Lemma 1** (Parker-Holder et al. (2020)). *Maximizing the final performance $F_T$ of a model with respect to a given hyperparameter schedule $\{\boldsymbol{h}_t\}_{t=1}^T$ is equivalent to maximizing the time-varying black-box function $f_t(\boldsymbol{h}_t)$ and minimizing the corresponding cumulative regret $\tilde{r}_t(\boldsymbol{h}_t)$:*

$$\max F_T(\boldsymbol{h}_T) = \max \sum_{t=1}^T f_t(\boldsymbol{h}_t) = \min \sum_{t=1}^T \tilde{r}_t(\boldsymbol{h}_t).$$

We find two issues with this lemma. The first is a minor technicality unrelated to our main argument: $\mathtt{max}$ and $\mathtt{min}$ should be $\mathtt{argmax}$ and $\mathtt{argmin}$. This is due to the proof (Eq. 3) using $F_T(\boldsymbol{h}_T) - F_1(\boldsymbol{h}_1)$ in place of $F_T(\boldsymbol{h}_T)$. Subtracting a constant (as $F_1(\boldsymbol{h}_1)$ is assumed to be) affects $\mathtt{max}$ but not $\mathtt{argmax}$.

Secondly, we find that the lemma is proved in the setting of non-sequential optimization (Eq. 1) rather than the sequential optimization actually performed by PBTs (Eq. 2). We show that $\tilde{r}$, the regret relative to the schedule $\{\tilde{\boldsymbol{h}}_t\}_{t=1}^T$, lacks meaning in the non-sequential setting, rendering any conclusions about it moot. Finally, we show that the lemma does not hold in the sequential setting. Let us start with the original proof:

$$\max\left[F_T(\boldsymbol{h}_T) - F_1(\boldsymbol{h}_1)\right] = \max \sum_{t=1}^T \left[F_t(\boldsymbol{h}_t) - F_{t-1}(\boldsymbol{h}_{t-1})\right] = \max \sum_{t=1}^T f_t(\boldsymbol{h}_t) = \min \sum_{t=1}^T \tilde{r}_t(\boldsymbol{h}_t). \tag{3}$$

Let us replace $\mathtt{max}$ with $\mathtt{argmax}$, specify the arguments, replace $F_1$ with $F_0$ to correct the indexing of the telescoping sum, and expand the last transition (in accordance to Parker-Holder et al. (2020)):

$$\begin{aligned}
\arg\max_{\{\boldsymbol{h}_t\}_{t=1}^T} \left[F_T(\boldsymbol{h}_T) - F_0(\boldsymbol{h}_0)\right] &= \arg\max_{\{\boldsymbol{h}_t\}_{t=1}^T} \sum_{t=1}^T \left[F_t(\boldsymbol{h}_t) - F_{t-1}(\boldsymbol{h}_{t-1})\right] \\
&= \arg\max_{\{\boldsymbol{h}_t\}_{t=1}^T} \sum_{t=1}^T f_t(\boldsymbol{h}_t) \stackrel{(4.3)}{=} \arg\max_{\{\boldsymbol{h}_t\}_{t=1}^T} \sum_{t=1}^T \left[f_t(\boldsymbol{h}_t) - f_t(\tilde{\boldsymbol{h}}_t)\right] = \arg\min_{\{\boldsymbol{h}_t\}_{t=1}^T} \sum_{t=1}^T \tilde{r}_t(\boldsymbol{h}_t).
\end{aligned} \tag{4}$$

Observe that due to the dependencies between the summands, the argument of $\mathtt{argmax}$ must be the entire schedule for the transitions in the original proof (Eq. 3) to hold as written. Thus, the lemma is proved in the non-sequential setting. Let us next consider transition (4.3) in Eq. 4: it holds because for each time $t$, $f_t(\tilde{\boldsymbol{h}}_t)$ is a constant (so these values do not influence $\mathtt{argmax}$). However, this holds not just for $\{\tilde{\boldsymbol{h}}_t\}_{t=1}^T$ but for any

schedule $\{\boldsymbol{h}_t^A\}_{t=1}^T \in H^T$: for each time $t$, $f_t(\boldsymbol{h}_t^A)$ is a constant, and therefore transition (4.3) will hold also with $f_t(\boldsymbol{h}_t^A)$ in place of $f_t(\tilde{\boldsymbol{h}}_t)$ on the right side. Therefore, maximizing $f_t$ is equivalent to minimizing the cumulative regret relative to this arbitrary schedule, $r_t^A(\boldsymbol{h}_t) = f_t(\boldsymbol{h}_t^A) - f_t(\boldsymbol{h}_t)$.

Thus, we can use the proof of the lemma to conclude that the regret relative to all possible schedules is minimized, making the original result about the specific regret $\tilde{r}$ lack meaning. The notion of minimizing regret *relative to a schedule* is problematic if the *final* performance is optimized *non-sequentially*: the optimal schedule will maximize $F_T$ regardless of the schedule used to define the regret, as seen in the transition (4.3). The lemma ignores the sequential nature of optimization that is fundamental to the design of PBTs and that is required for a meaningful notion of regret relative to a schedule.

Next, we give a high-level argument that the lemma does not hold in the sequential setting, with details provided in Appendix A. $f_t(\boldsymbol{h}_t)$ is defined as $F_t(\boldsymbol{h}_t) - F_{t-1}(\boldsymbol{h}_{t-1})$, which reduces to $F_t(\boldsymbol{h}_t)$ during sequential optimization because $F_{t-1}(\boldsymbol{h}_{t-1})$ is constant at time $t$. With $f_t(\boldsymbol{h}_t) = F_t(\boldsymbol{h}_t)$, performance at the current step is maximized rather than the final performance. Due to potential dependencies on the past choices, greedily optimizing $F_t$ leads to worse final results (unless no dependencies are present, which would make the greedy solution optimal). Thus, contrary to the statement of Lemma 1, sequentially maximizing $f_t$ is *not* equivalent to maximizing the final performance $F_T$.

In the theory underlying all Bayesian PBTs, Lemma 1 connects minimizing regret to achieving optimal final performance. The associated regret bounds still hold, and it is true that the returns of the schedules discovered by Bayesian PBTs asymptotically approach those of $\{\tilde{\boldsymbol{h}}_t\}_{t=1}^T$, as shown by Parker-Holder et al. (2020). Our contribution lies in reinterpreting these results: the returns achieved by the Bayesian PBTs asymptotically approach the returns of the *greedy* solution (which we find $\{\tilde{\boldsymbol{h}}_t\}_{t=1}^T$ to be) that will only be optimal if previous hyperparameter choices do not restrict achievable future results (which cannot be generally expected).

## 4.2 Limitations of the current theoretical approach

The theoretical asymptotic behavior, while informative, does not completely describe an algorithm's behaviour in practice. Since Bayesian PBTs work with a population, a solution that does not lead to the best performance at the current step may still survive until the next step, where it may actually perform the best. Together with the search space exploration performed by BO, this enables Bayesian PBTs to discover schedules different from the greediest one (note that the `exploit` procedure can also influence the (asymptotic) behavior despite not being included in the theory of Bayesian PBTs, see Appendix B).

Nonetheless, Bayesian PBTs are solving an optimization problem that is fundamentally not aligned with the stated goal of optimizing the final performance. The TV-GPB formalization is well-suited for time-varying problems where previous choices do not affect future results, such as adjusting a thermostat (Bogunovic et al., 2016). However, dynamic HPO typically does not fit this criterion, as it is a time-linked problem (Bosman, 2005), meaning that previous choices can strongly influence downstream results: e.g., reducing the learning rate too fast may improve performance in the short term, but in the end leads to premature convergence and a suboptimal final result.

Note that the standard PBT optimizes the same greedy objective function as Bayesian PBTs, but it does so less effectively due to its simple exploration via random perturbation (in contrast to BO). However, since the greedy objective function being optimized does not completely align with the true objective function (the final performance), the less effective optimization of PBT can lead to better final results.

FIRE-PBT stands out because its higher-order subpopulations greedily optimize not the current performance but the improvement rate, which encourages better long-term results. Changing the optimization problem is one way of aligning an online algorithm with the target objective, see Section 6 for further discussion.

## 4.3 Step size as a mechanistic cause for the greediness of the PBT variants

All PBT variants operate on two levels. As discussed in Sections 2.1 and 3, on the upper level, a PBT performs $T$ outer `steps` of HPO, while on the lower level, within each outer `step`, the underlying models are

trained for $\text{step}_{\text{inner}}$ gradient descent (GD) steps, to the total of $\text{total}_{\text{inner}} = T \cdot \text{step}_{\text{inner}}$ steps over the entire run. While the $\text{total}_{\text{inner}}$ budget is fixed, the number of outer `steps` $T$ is a hyperparameter of PBT that can be set freely by appropriately adjusting $\text{step}_{\text{inner}}$: e.g., if the training budget is 100 epochs, they can be split into 100 outer `steps` of 1 epoch, or 10 outer `steps` of 10 epochs, etc.

PBT variants optimize the objective function after one outer `step`, which can correspond to an arbitrary number of inner steps. PBTs are blind to how long the inner optimization is: no matter if the weights are trained for a single GD step or for many epochs, it is still a single outer `step` for a PBT. What happens if we increase the number of outer steps $T$? Holding $\text{total}_{\text{inner}}$ constant, this requires decreasing $\text{step}_{\text{inner}}$. A PBT will still optimize one outer `step` ahead but fewer inner steps ahead. In terms of inner steps, the optimization becomes greedier, as shorter-term improvements are pursued.

Thus, one mechanistic cause for how greedy a specific PBT variant behaves is the number of outer steps $T$ (or, equivalently, $\text{step}_{\text{inner}}$). It is not clear how $T$ should be set, as none of the PBT publications discuss it, while effectively using different values: 200 in PBT, 20 in PB2, 50 in PB2-Mix, 150 in BG-PBT.

Decreasing $T$ could decrease greediness, as it is equivalent to increasing $\text{step}_{\text{inner}}$, i.e., how many inner steps ahead the optimization is done. However, this would also make the schedule less granular and reduce the amount of exploration an algorithm can do, which could degrade performance in challenging search spaces.

The optimal amount of greediness and thus the optimal value of $T$ (or $\text{step}_{\text{inner}}$) is likely to be task-specific, and may even vary during the optimization: Wan et al. (2022) empirically observe that their chosen $\text{step}_{\text{inner}}$ leads to excessively greedy behavior on some tasks, and add linear annealing of $\text{step}_{\text{inner}}$ for these tasks. We argue that $\text{step}_{\text{inner}}$ is an important hyperparameter which could influence the behavior of PBTs in *any* task. We empirically demonstrate in Section 5.4 that varying the number of outer `steps` can change not only the final performance but also which PBT variant performs best.

## 5 Experiments & Results

### 5.1 General setup

We compare the PBT variants on image classification, RL, and toy problems. In our main classification experiments, the PBTs optimize hyperparameters of a ResNet-12 (He et al., 2016) on Fashion-MNIST (Xiao et al., 2017) and CIFAR-10 (Krizhevsky & Hinton, 2009) datasets (see Appendix C for further details).

In main RL experiments, a Proximal Policy Optimization (PPO) (Schulman et al., 2017) agent is trained on the Hopper and Humanoid tasks from the Brax library (Freeman et al., 2021). These tasks were selected because Wan et al. (2022) reported that PBTs tended to get stuck in local optima if the step size was small and we wanted to see if Perturbation PBTs could avoid local optima better than Bayesian PBTs.

By default, only the Learning Rate (LR) is tuned in the classification experiments, i.e., the *small* search space is used (see Appendix E for the description of search spaces) but we found that for the Humanoid task this leads to minimal performance differences between PBTs (see Section 5.5) which motivated using the *large* search space in the RL experiments. For classification, the *large* search space is used in the scaling experiments (Sections 5.6 and 5.7) so that the scaling of PB2-Mix could be observed. PB2-Mix is equivalent to PB2 in continuous search spaces and so is excluded from the experiments in the *small* space.

Unless specified otherwise, the PBTs are run with a population of size 22, see Appendix D for the explanation, as well as other implementation details of PBTs. Five random seeds were used for the classification and toy experiments, while seven seeds were used for RL (known to have high variability). We typically plot Interquartile Mean (IQM) and Interquartile Range (IQR) in order to focus on average tendencies rather than outliers, similar to previous work (Parker-Holder et al., 2020). Statistical testing is described in Appendix F.

### 5.2 Toy problems: plain and time-linked

In this section, we describe two toy problems: one where effective greedy optimization leads to better results, and another where the less greedy optimization is preferable.

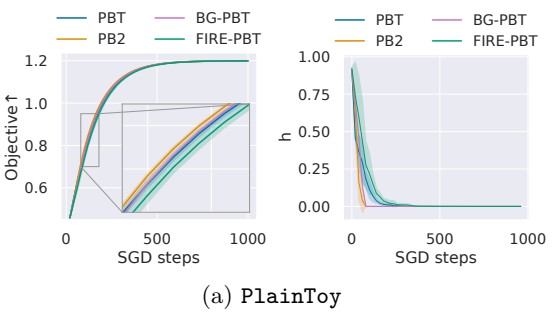
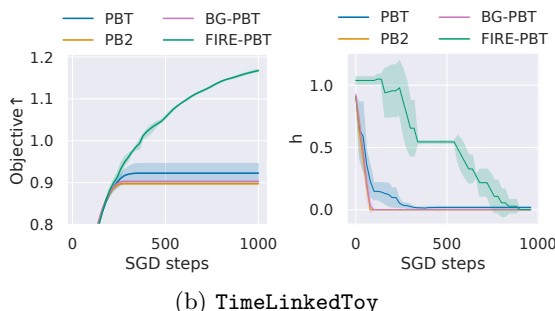

(a) `PlainToy`          (b) `TimeLinkedToy`

Figure 2: The performance of PBTs on the toy problems. For each problem, on the left, the IQM of the objective is shown (with IQR shaded), and on the right, the average value of the hyperparameter $h$ (with the standard deviation shaded).

The problems are created based on the toy problem from Jaderberg et al. (2017), where a quadratic function $Q(\theta) = 1.2 - (\theta_0^2 + \theta_1^2)$ is maximized without knowing its formula. Instead, a differentiable surrogate function is provided $\hat{Q}(\theta) = 1.2 - (h_0\theta_0^2 + h_1\theta_1^2)$, with $h = [h_0, h_1]$ as the hyperparameters to be optimized. The gradient is $\nabla_\theta \hat{Q} = -2h\theta$, therefore $h$ should be greedily maximized to reach the optimum faster. We slightly adjust this problem and refer to it as `PlainToy`. Namely, we make $\theta$ one-dimensional, and replace $h$ with $2 - h$ to prevent $h$ from going to infinity (while allowing it to increase up to 2). The objective is $g_1(\theta) = 1.2 - \theta^2$, the surrogate is $\hat{g}_1(\theta) = 1.2 - (2 - h)\theta^2$, the gradient is $\nabla \hat{g}_1 = -2(2 - h)\theta$. The initial values are uniformly randomly sampled from $[0.9, 1.1]$. This problem is designed so that the greedy behavior of reducing $h$ to 0 as fast as possible is also the optimal behavior that leads to the best final result. On this problem, the greedier algorithms are expected to outperform the less-greedy ones.

Our second problem, `TimeLinkedToy`, is designed so that the greediest solution is optimal in the short-term but suboptimal in the long-term. It is an extension of the `PlainToy` with an additional penalty that is incurred by deviating from a linear decay schedule. As such, $2 - h$ is replaced with $max(2 - h - c \cdot p_{t-1}, 0)$, where $p_{t-1} = \sum_{i=0}^{t-1} |h^i - \frac{T-i}{T}|$ ($h^i$ denotes the historical value of $h$ at time $i$) and $c = 0.2$ (a constant to control the impact of the penalty). The "max" function is added to prevent the surrogate gradient from degrading the objective when the penalty grows large (making the gradient incorrect). The objective is $g_2(\theta) = 1.2 - \theta^2$, the surrogate is $\hat{g}_2(\theta) = 1.2 - \max(2 - h - c \cdot p_{t-1}, 0) \cdot \theta^2$, the gradient is $\nabla \hat{g}_2 = -2 \cdot max(2 - h - c \cdot p_{t-1}, 0) \cdot \theta$. With $c < 1$, rapidly reducing $h$ gives the best results in the first steps but the penalty gradually adds up, eventually stopping all progress.

### 5.3 Bayesian PBTs underperform Perturbation PBTs when greedy choices are harmful in the long term

Based on our theoretical analysis in Section 4, Bayesian PBTs should perform worse than Perturbation PBTs when short-term improvement is detrimental to long-term success (and they should perform better if not). The goal of the experiments in this section is to empirically validate these claims.

**Toy problems** We start with the `PlainToy` problem, where the greediest solution is optimal. Figure 2a shows that all PBT variants successfully solve `PlainToy` by reducing the hyperparameter $h$ to zero. Bayesian PBTs find the optimum faster than PBT, achieving better intermediate results. The less greedy FIRE-PBT reduces $h$ the slowest, which leads to the slowest improvement (see the inset in Figure 2a (left)).

The results are different for the `TimeLinkedToy` problem (Figure 2b), where the greediest solution is optimal in the short but not the long term. Bayesian PBTs again quickly reduce $h$ to zero, which leads to large penalties downstream due to deviating from the target linear decay schedule. Cumulative penalties eventually stop all progress of Bayesian PBTs, leading to poor final performance. Standard PBT, which cannot reduce the LR as quickly, performs better than the Bayesian PBTs, while FIRE-PBT achieves the best result by successfully addressing time linkage.

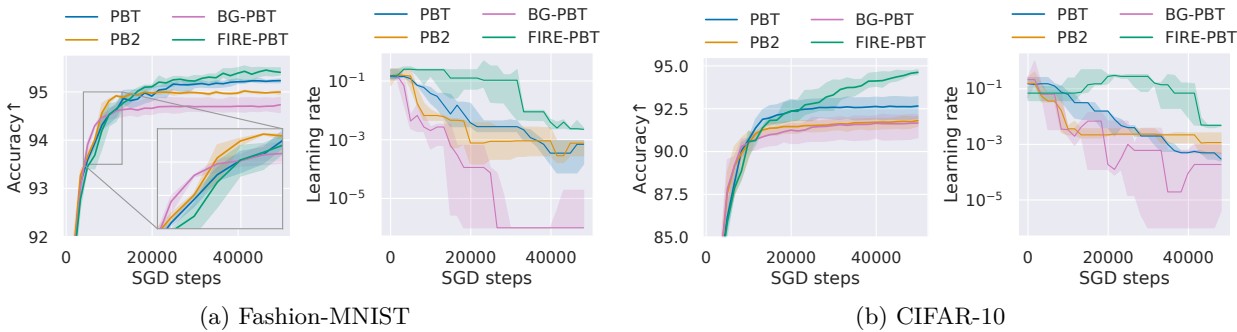

Figure 3: The performance of PBTs on the classification tasks. For each dataset, on the left, the IQM of the accuracy is shown, and on the right, the median LR of the best solution. The IQRs are shaded.

**Classification**    Figure 3 shows the results of optimizing the LR of an image classifier. For both datasets, Perturbation PBTs on average outperform Bayesian PBTs in terms of final accuracy. This is due to Bayesian PBTs decaying the LR too aggressively. It is interesting to see in Figure 3a that the latest PBT variant, BG-PBT, reduces LR the fastest and on average performs better at the start of the training — only to be later overtaken by a less recent PB2 that reduces the LR somewhat slower. PB2, in its turn, is overtaken by PBT that reduces the LR even more gradually. However, by that point, the eventual winner, FIRE-PBT, already outperforms others despite starting off as the worst. Figure 3b shows similar behaviour.

These results are consistent with our analysis in Section 4. They clearly demonstrate in a practical setting that *more effective* Bayesian optimization can, in fact, be *greedier* optimization, ultimately leading to worse results. Note, however, that depending on the budget, different algorithms should be considered best: for Fashion-MNIST, BG-PBT performs best after ≈ 5,000 steps, whereas PB2 is the best after ≈ 10,000 steps. This implies that the properties of the task as well as the budget influence which PBT should be used for obtaining the best results.

**Reinforcement learning**    Figure 4 shows the scores of the PBTs on RL tasks when either 30 or 100 outer steps are used. In the latter case, the algorithm with the highest final score on both tasks is a Bayesian PBT (BG-PBT for Hopper, PB2-Mix for Humanoid), performing well throughout the training. Although a pattern of rapid improvement followed by stagnation is visible in Figure 4b (similar to the classification tasks), Perturbation PBTs generally fail to overtake the Bayesian ones like they did in the classification tasks. When 30 outer steps are used (Figure 4a), Bayesian PBTs again tend to be better, although PBT marginally outperforms BG-PBT on Hopper (while continuing to perform poorly on Humanoid). We can conclude that in complex RL tasks and search spaces, the greedier but more effective optimization of Bayesian PBTs can lead to better results than the less greedy and less effective optimization of Perturbation PBTs.

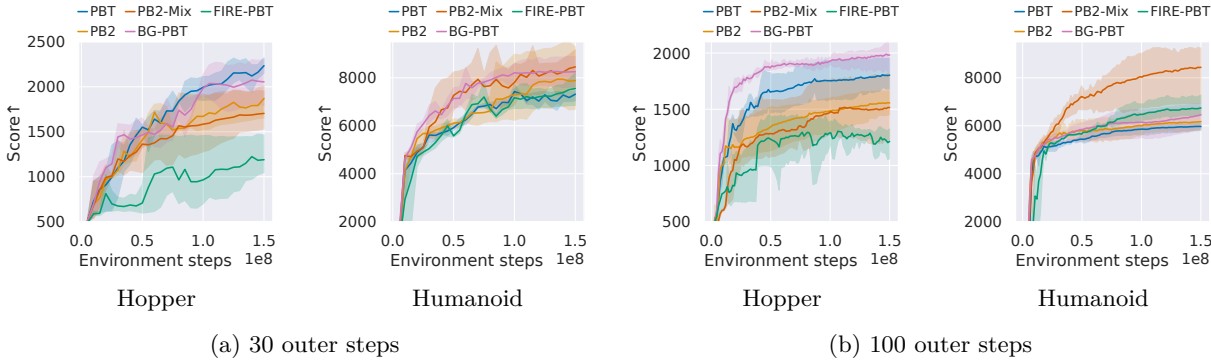

Figure 4: The performance (IQM, with IQR shaded) of PBTs on the RL tasks with 30 or 100 outer `steps`.

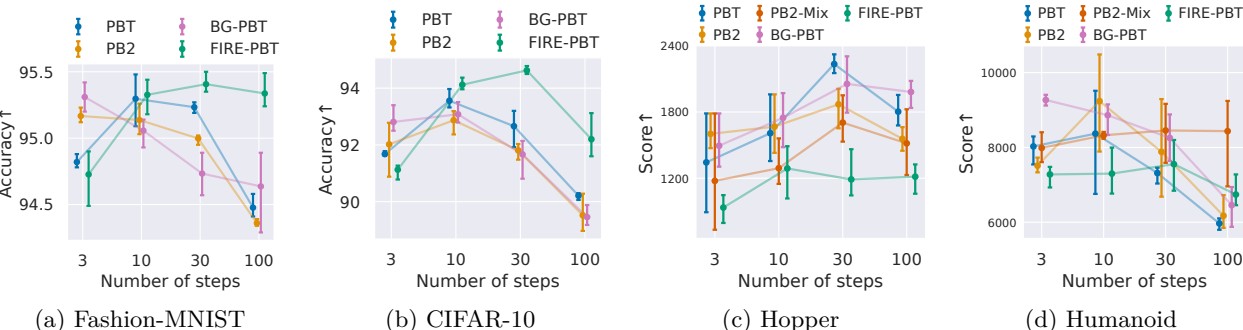

Figure 5: The performance (IQM and IQR) of the PBT variants as the number of outer `steps` is varied, while the number of inner steps (Stochastic Gradient Descent (SGD) steps for classification, environment steps for RL) is kept constant.

## 5.4 The number of outer steps impacts the greediness and the relative performance of PBT variants

In Section 4.3, we reasoned that increasing the number of outer `steps` (while keeping the number of inner steps constant) should make a PBT variant greedier. Let us first examine the results for the classification tasks in Figure 5 (a, b). Bayesian PBTs perform well when the number of outer `steps` is very low (3) as they are able to effectively find good LR values despite having few exploration opportunities, unlike the less-effective Perturbation PBTs (which consequently perform worse in this setting). However, as the number of outer `steps` increases, the performance of Bayesian PBTs tends to worsen, since they start to effectively optimize the increasingly shorter-term objective.

On the other hand, Perturbation PBTs strongly improve when the number of outer `steps` goes from 3 to 10, as they have more opportunities to explore the search space but not enough to arrive at the greedy solution that optimizes short-term performance improvement. However, as the number of outer `steps` is further increased, the Perturbation PBTs become capable of effective greedy optimization, and their performance deteriorates (at 30 `steps` for PBT, at 100 `steps` for FIRE-PBT).

The impact of the number of outer `steps` on RL scores (Figure 5 (c, d)) does not follow a clear pattern. PBT peaks at different number of outer `steps` in different tasks while FIRE-PBT show consistently poor performance. BG-PBT tends to outperform PB2 and PB2-Mix but as the number of outer `steps` is increased, its performance improves on Hopper and deteriorates on Humanoid. As discussed in Section 5.3, greedy optimization does not appear as detrimental for these tasks as for the classification tasks, explaining absence of a similar interpretable pattern. Nonetheless, the number of outer `steps` affects both the absolute and the relative performance of the PBT variants (in most cases), highlighting the importance of this HP also in tasks without strong time linkage.

## 5.5 As the search space size is increased, performance of PBTs tends to improve

We investigated how the relative and absolute performance of PBT variants change when the size of the search space is varied. Figure 6 shows the performance of PBTs in search spaces of different sizes (note that search spaces differ between the classification and RL tasks, see Appendix E).

As can be seen in Figure 6 (a, b), for classification, FIRE-PBT tends to perform best in all search spaces, with performance increasing in larger search spaces (which shows that optimizing additional hyperparameters is beneficial). The performance of PBT improves when going from the *small* to the *medium* search space but deteriorates in the *large* space. The latter observation is likely due to inability of PBT to optimize the increased number of variables. The performance of Bayesian PBTs tends to increase with the search space size, although without reaching the level of performance of FIRE-PBT.

Figure 6 (c, d) shows that the performance of all PBT variants improves on the RL tasks as more HPs are optimized in larger search spaces. The relative performance of PBTs varies strongly, with no algorithm

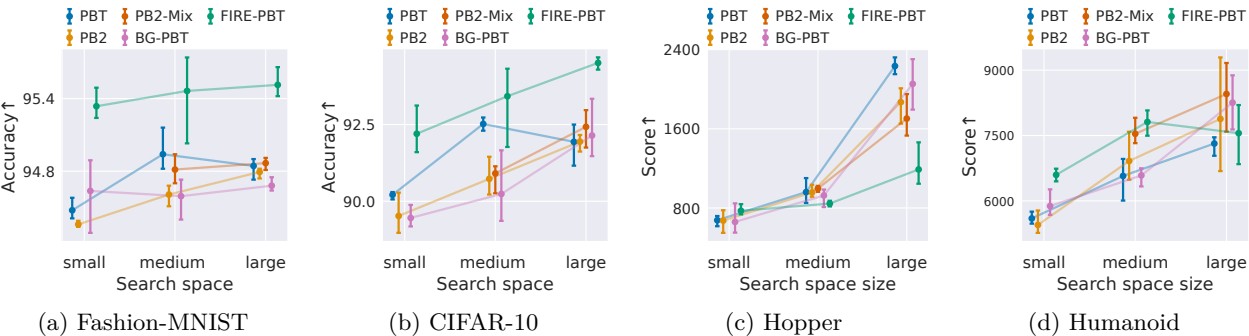

(a) Fashion-MNIST  (b) CIFAR-10  (c) Hopper  (d) Humanoid

Figure 6: The performance (IQM and IQR) of the PBT variants as the search space size is increased. Some IQR marks are invisible due to the IQR being too narrow.

significantly outperforming others or even consistently achieving a higher average rank (see Appendix F). Interestingly, the difference in performance of the same PBT variant *across* search spaces can be larger than the differences between PBTs in the *same* search space. The disappointing relative performance of FIRE-PBT in most RL search spaces despite its excellent performance across classification search spaces serves as evidence that FIRE-PBT is not generally superior to the other PBT variants.

### 5.6 As the population size is increased, performance of PBTs tends to improve

We investigated the effect of increasing the population size on the performance of the PBT variants. Bayesian PBTs were designed to work well with a small population (typically 8, although scaling to larger populations was shown to be beneficial) while Perturbation PBTs tend to use a population size of at least 20.

Consequently, even for the time-linked classification tasks, PBT and FIRE-PBT do not outperform the Bayesian PBTs if the population size is 8 (see Figure 7 (a, b)). However, as the population size increases, FIRE-PBT strongly outperforms all the other variants on the classification tasks, with PBT tending to be second-best as the population size reaches 50.

Figure 7 (c, d) shows that on RL tasks, Perturbation PBTs again perform poorly with a population size of 8. While they tend to improve with an increasing population size, they typically still underperform Bayesian PBTs, except for PBT on Hopper. PBT performs well not only with a population size of 22 (as previously seen in Section 5.3), but with a population size of 50 as well. Which properties of the task lead to these results is an open question. In general, performance of all PBTs tends to improve as the population size is increased, consistently with their nature as population-based algorithms. At the same time, the relative positions of the PBT variants changes across population sizes and tasks, with no algorithm significantly outperforming others, except for FIRE-PBT on classification tasks with population sizes of 22 and 50 (see Appendix F).

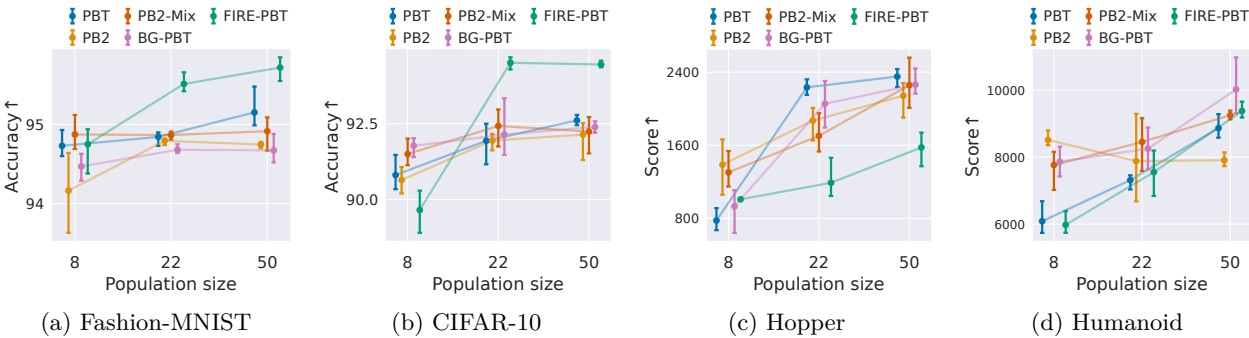

(a) Fashion-MNIST  (b) CIFAR-10  (c) Hopper  (d) Humanoid

Figure 7: The performance (IQM and IQR) of the PBT variants as the population size is increased.

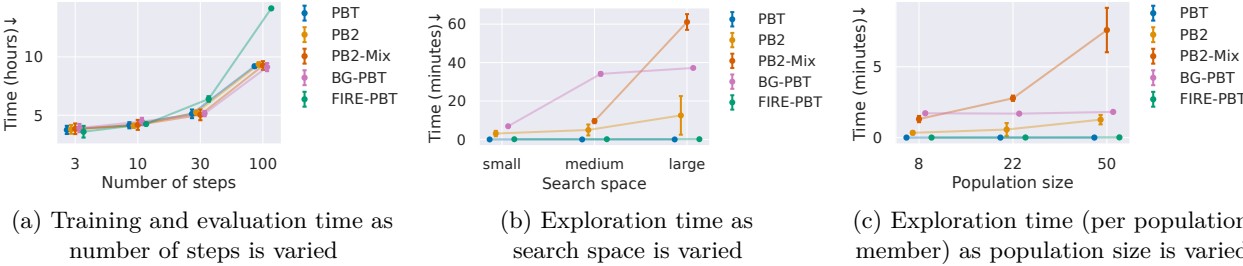

(a) Training and evaluation time as number of steps is varied

(b) Exploration time as search space is varied

(c) Exploration time (per population member) as population size is varied

Figure 8: Wall-clock time (mean $\pm$ st. dev.) taken by the PBTs on the Hopper task in various setups.

## 5.7 PBTs scale differently in terms of wall-clock time

An attractive property of PBT variants is their ability to optimize hyperparameters within the wall-clock time of a single training run, if all $N$ networks in the population are trained in parallel. We would like to add nuance by considering how much extra time the advanced PBT variants spend, and what they spend it on.

The first source of extra time spending is intermediate evaluations. At the end of each outer `step`, all networks need to be evaluated, with more outer steps requiring more evaluations and longer wall-clock time.

All PBT variants except for FIRE-PBT use the same number of evaluations. FIRE-PBT requires an order of magnitude more intermediate evaluations to compute the IR objective used in the higher-order subpopulations. As a result, FIRE-PBT tends to take the longest wall-clock time, which is noticeable for RL tasks where evaluations interrupt efficient Brax-enabled training. Figure 8 (a) shows that increasing the number of steps increases the total time spent on training and evaluation by all the algorithms but especially by FIRE-PBT.

The second source of extra time spending comes from selecting new hyperparameters in the `explore` step. The Bayesian PBTs optimize the hyperparameters of the GP and the acquisition function, while the random perturbation of PBT is near-instantaneous. FIRE-PBT fits several GPs when computing the IR objective. The cost of these additional operations could be influenced by the search space and the population size.

As can be seen in Figure 8 (b), increasing the search space size strongly affects only BG-PBT and PB2-Mix. BG-PBT slows down considerably when going from a continuous search space (*small*) to a mixed search space (*medium*), but does not meaningfully slow down when the number of variables is further increased in *large*. PB2-Mix, on the other hand, is slower in the *large* search space than in the *medium* search space. This is consistent with BG-PBT relying on a more efficient BO algorithm. Figure 8 (c) shows that increasing the population size does not affect per-population-member update times, except for PB2-Mix: a larger population creates more data for the fitting of a GP, which appears to slow down more than linearly (which would correspond to a horizontal line). The exploration of FIRE-PBT incurs negligible costs, as does the exploration of PBT.

## 5.8 Do our observations generalize to similar settings?

Our computation budget is limited (since each PBT run entails training multiple neural networks, the total computational cost of the main experiments reach $\approx 900$ GPU-days) but we would still like to know whether changes in the experimental setup would substantially alter the results. For this purpose, we run additional experiments in selected settings.

**Classification** CIFAR-100 (Krizhevsky & Hinton, 2009) is a step-up in complexity from CIFAR-10 in number of classes. Tiny ImageNet (Stanford CS231N, 2017) is larger still in number of classes, number of samples, and resolution (which we upsample to $224^2$). This motivates training a larger architecture (ConvNeXt-T (Liu et al., 2022)) for 3 times more steps. We use the *large* search space, and 100 outer `steps`, further details are provided in Appendix C.

Figure 9a shows that FIRE-PBT performs best on CIFAR-100, as expected. The standard PBT does not outperform the Bayesian PBTs, likely due to the search space being large, which is consistent with our

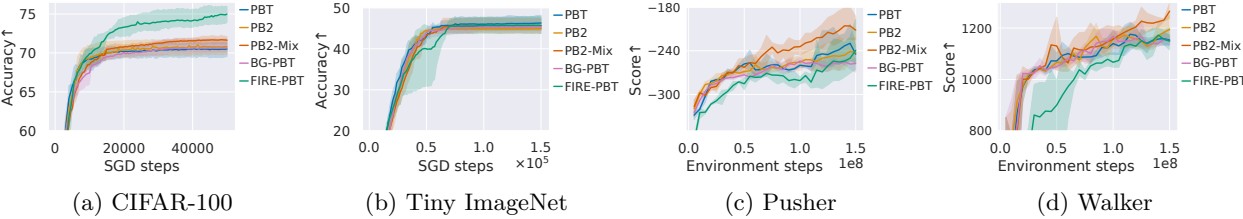

Figure 9: The performance (IQM, with IQR shaded) of PBTs on additional tasks.

findings presented in Section 5.5. On Tiny ImageNet, however, FIRE-PBT manages only to catch up to other PBTs but is not able to overtake them. The combination of increased difficulty of the underlying task and the large search space may strain the simple exploration procedure of FIRE-PBT that nonetheless performs best in the *small* space, as expected (see Appendix G). These results highlight the difficulty in choosing the PBT variant that would perform best for a specific task, search space, and budget, as discussed in Section 5.3.

**Reinforcement learning**  We chose to evaluate on Brax Pusher and Walker tasks because they were not previously optimized by any PBT variant. Figure 9 shows that the algorithm that achieves the best final result, PB2-Mix, tends to perform well throughout optimization, with no issues caused by greediness (similarly to what we observed on Hopper and Humanoid). PB2-Mix outperforming BG-PBT on both tasks is consistent with our previous results, where PB2-Mix was also found to be better than BG-PBT in some settings. On the other hand, in some previously considered settings, BG-PBT tended to perform better, preventing the conclusion that one of the two algorithms is superior.

## 6   Discussion & Conclusion

In this work, we compared five single-objective PBT variants on image classification and RL tasks, and found no single best algorithm that is capable of outperforming others in all (or even most) considered settings. This is partially explained by different PBT variants exhibiting (and different tasks requiring) different degrees of greediness. If good long-term results are achieved by optimizing short-term performance gains, then effective optimization of Bayesian PBTs is beneficial. If, however, effectively optimizing short-term gains compromises long term results, then even the standard PBT can outperform Bayesian PBTs, with FIRE-PBT performing even better. In our experiments, optimizing RL hyperparameters fell into the first category, while optimizing the LR of an image classifier typically fell into the second one.

From a theoretical perspective (as discussed in Sections 4.1 and 4.2), the underwhelming results of BO in time-linked settings stem from effectively solving an optimization problem that is not fully aligned with the desired goal of maximizing final performance. Is there an alternative to formalizing dynamic HPO as optimization of a time-varying function? In Eq. 1, we defined as optimum the schedule optimized over all steps simultaneously, which is incompatible with the efficiency brought by the sequential optimization of PBTs. Nonetheless, greedy optimization could be improved upon, e.g., by using an estimate of the future performance as the objective (Bosman, 2005), or, as in FIRE-PBT, by letting a subset of the population optimize an alternative objective directed at long-term performance.

The objective of all PBT variants except FIRE-PBT is to optimize performance after one outer `step` of the algorithm. We observe that by changing how much inner optimization (e.g., via gradient descent) is performed within an outer `step`, we can influence how greedily an algorithm behaves on time-linked tasks (although it is not the only effect, as discussed in Section 4.3). Varying the step size can substantially affect both the absolute and the relative performance of the PBT variants in all settings, with no single best value across algorithms and settings. Potentially having to try several values reduces the benefits of PBTs: efficiency and running within the wall-clock time of a single training. In order for a more complete picture of the upsides and downsides of different approaches to dynamic HPO to emerge, alternative dynamic HPO algorithms should be studied and compared both to PBTs and to each other.

PBT variants typically do not add much wall-clock time to the training of the underlying model (if it is large enough), with the exception of FIRE-PBT. Its success in optimizing time-linked problems comes at a cost of an order of magnitude higher number of evaluations, which can be ignored if the evaluations are fast but adds up if they are not (as in our RL experiments). Future work could investigate the possibility of performing fewer intermediate evaluations or formulating a more efficient alternative to the improvement rate objective.

While we separately varied the number of steps of PBTs, their population sizes, and the search-space sizes, we did not tune them jointly, nor have we fully explored all PBT hyperparameters (e.g., we varied $\lambda$ of the truncation selection only in a few settings, as reported in Appendix I; the perturbation factor of FIRE-PBT was also varied in a limited number of settings, see Appendix J). Arguably, efficient HPO algorithms should perform well without requiring tuning of their HPs, as this only shifts the HPO problem to a higher level. While the untuned hyperparameters is a limitation of our work, we believe that our conclusions on the interplay of the greediness and performance of the PBT variants in time-linked settings are general, given their theoretical underpinnings that align with empirical results. Deeper investigation of the interaction between the hyperparameters, the performance, and the behaviour of PBTs is a potentially interesting subject for future work.

Although our theoretical and empirical results can explain behaviour of PBT variants optimizing the LR of a classifier, they are far from a complete explanation of behaviour of PBTs on arbitrary tasks. Further understanding why a PBT variant performs poorly or well on a specific underlying task is an intriguing avenue of future research.

### Acknowledgments

We thank the anonymous reviewers for their helpful feedback. This work is part of the research project DAEDALUS which is funded via the Open Technology Programme of the Dutch Research Council (NWO), project number 18373; part of the funding is provided by Elekta and ORTEC LogiqCare.

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

## A    Detailed reasoning on Lemma 1 in sequential setting

Here we show that Lemma 1 does not hold in the sequential setting: specifically, that sequential maximization of $f_t$ does not generally result in optimal $F_T$. Parker-Holder et al. (2020) consider as optimal the schedule $\{\tilde{\boldsymbol{h}}_t\}_{t=1}^{T} = \{\arg\max_{\boldsymbol{h}_1} f_1(\boldsymbol{h}_1), \arg\max_{\boldsymbol{h}_2} f_2(\boldsymbol{h}_2), \ldots, \arg\max_{\boldsymbol{h}_T} f_T(\boldsymbol{h}_T)\}$. Recall that $f_t(\boldsymbol{h}_t)$ is defined as $F_t(\boldsymbol{h}_t) - F_{t-1}(\boldsymbol{h}_{t-1})$. For sequential optimization, $F_{t-1}(\boldsymbol{h}_{t-1})$ is constant at time $t$. Therefore, it does not influence the optimization of $f_t(\boldsymbol{h}_t)$, allowing us to replace $\arg\max_{\boldsymbol{h}_t} f_t(\boldsymbol{h}_t)$ with $\arg\max_{\boldsymbol{h}_t} F_t(\boldsymbol{h}_t)$, revealing that $\{\tilde{\boldsymbol{h}}_t\}_{t=1}^{T}$ is simply the greedy schedule $\{\arg\max_{\boldsymbol{h}_1} F_1(\boldsymbol{h}_1), \arg\max_{\boldsymbol{h}_2} F_2(\boldsymbol{h}_2), \ldots, \arg\max_{\boldsymbol{h}_T} F_T(\boldsymbol{h}_T)\}$. Consequently, sequentially maximizing $f_t$ is equivalent to greedily maximizing $F_t$ at each $t$ (in other words, optimizing a proxy $f_t$ (rather than the objective itself, $F_t$) makes no difference in the outcome).

Such greedy maximization does not, in general, maximize the final performance. Although we and Parker-Holder et al. (2020) write $F_t$ as a function of only $\boldsymbol{h}_t$, this is not correct in the context of dynamic HPO. Previous hyperparameter choices can influence future results via the weights inherited from the previous step (e.g., too high a learning rate can lead to divergence that cannot be recovered from). Thus, performance at step $t$ is not generally independent of the previous steps, which can be made explicit: $F_t(\boldsymbol{h}_t|\boldsymbol{h}_1 \ldots \boldsymbol{h}_{t-1}) = Q(\texttt{train}(\theta_1|\{\boldsymbol{h}_i\}_{i=1}^{t}))$.

Due to potential dependencies on the past choices, greedily optimizing $F_t$ leads to worse results (unless no dependencies are present, which would make the greedy solution optimal):

$$\max_{\{\boldsymbol{h}_t\}_{t=1}^T} F_T(\boldsymbol{h}_T|\boldsymbol{h}_1\dots\boldsymbol{h}_{T-1}) \geq F_T(\tilde{\boldsymbol{h}}_T|\tilde{\boldsymbol{h}}_1\dots\tilde{\boldsymbol{h}}_{T-1})$$

$$\text{where } \{\tilde{\boldsymbol{h}}_t\}_{t=1}^T = \{\arg\max_{\boldsymbol{h}_1} F_1(\boldsymbol{h}_1), \arg\max_{\boldsymbol{h}_2} F_2(\boldsymbol{h}_2|\tilde{\boldsymbol{h}}_1), \dots, \arg\max_{\boldsymbol{h}_T} F_T(\boldsymbol{h}_T|\tilde{\boldsymbol{h}},\dots\tilde{\boldsymbol{h}}_{T-1})\}.$$

To sum up, in the sequential setting, contrary to the statement of Lemma 1, maximizing $f_t$ is *not* equivalent to maximizing the final performance $F_T$ (even though it is equivalent to minimizing regret $\tilde{r}_t$).

## B    Theoretical consequences of the `exploit` procedure: an example

One can imagine a variant of the truncation selection where only the best solution in the population survives, and all others are replaced, with the weights copied from this solution, and hyperparameters explored via BO. Asymptotically, as population size goes to infinity, the best greedy solution maximizing current performance will be discovered at each step $t$. Only the weights from this solution will be present in the population of the next step, thus making sure that all schedules discovered in the future have the "greedy" hyperparameters at the time $t$. The final schedule will have the "greedy" hyperparameters at each step, i.e., it will be the greediest schedule.

On the other hand, if truncation selection with $\lambda < 50\%$ is used, the population size going to infinity allows for many suboptimal-in-the-short-term weights to be preserved at each step, likely preventing the dominance of the greediest solution.

This is only an illustrative example of the potential influence of the `exploit` procedure on the results that can occur despite the BO component remaining constant. We hope to encourage a deeper theoretical treatment of the `exploit` procedure in future research.

## C    Task implementation details

**Classification**    The default training hyperparameters are: $50,000$ SGD steps with a batch size of 128, a learning rate of 0.1, weight decay set to 0.0005, and the use of RandAugment (Cubuk et al., 2020) with 1 augmentation of magnitude 10.

The CIFAR-10/100 (Krizhevsky & Hinton, 2009) datasets contain 50,000 training and 10,000 testing samples each, we further separate 10,000 images from the training set to use as the validation set. The Fashion-MNIST (Xiao et al., 2017) contains 60,000 training and 10,000 testing samples, we further separate 10,000 images from the training set to use as the validation set. The Tiny ImageNet (Stanford CS231N, 2017) dataset contains 100,000 training, 10,000 validation, and 10,000 test samples. The labels for the test set are not available, therefore we treat the original validation set as the test set and separate 10,000 images from the training set to use as the validation set.

For each dataset, the validation set is used during the search to estimate the quality of a solution. The results in the main text are reported on the validation set so that the algorithms would be compared in terms of the objective they were optimizing. It also allows us to compare the intermediate performance. The final models are also evaluated on the test set. We observe that the results are similar to those on the validation set, see Figures 12-14 (Appendix H).

**RL**    For the RL tasks, we follow the setup of Wan et al. (2022) where hyperparameters of a PPO (Schulman et al., 2017) agent are tuned for 150 million environment interactions on the tasks from the Brax library (Freeman et al., 2021). To avoid overfitting to a single seed, a different random seed is set in each outer `step` of a PBT. The test performance is evaluated on a previously unseen seed. Both the validation and the test rewards are averages over 256 parallel rollouts. The test results are presented in Figures 12-14 (Appendix H). We would like to highlight that the validation and test performance on the RL tasks is almost identical. The common RL issue of the train-test performance gap (Eimer et al., 2023) was avoided by using a different random seed in each outer step of a PBT algorithm.

For further implementation details, see our codebase: `https://github.com/AwesomeLemon/PBT-Zoo`. The codebase includes configuration files for all experiments, as well as code for plotting and statistical testing.

## D   PBT variants implementation details

Population sizes are set based on the requirements of FIRE-PBT: while the other PBTs have a single population, we use FIRE-PBT with two hierarchical subpopulations and an evaluator subpopulation. To avoid the evaluators staying idle, their number should be equal to $(100 - \lambda)\%$ of the second subpopulation, where $\lambda$ is the hyperparameter of the truncation selection (25% by default). We follow Dalibard & Jaderberg (2021) in using population sizes of 22 (two subpopulations of size 8, and 6 evaluators) and 50 (two subpopulations of size 18, and 14 evaluators). We additionally consider the smaller population size of 8, which is typically used by the Bayesian PBTs. In this setting, FIRE-PBT has two subpopulations of size 3, and 2 evaluators. The $\lambda$ for FIRE-PBT is appropriately increased to 34% from the default 25%.

In Perturbation PBTs, we use $\{0.5, 2.0\}$ as factors for perturbation. For some hyperparameters with narrow ranges, a perturbation can only flip between the maximum and minimum values, preventing reasonable exploration. To address this, in our implementation, we check if the variable range is narrow like this, and if so, normalize the range to $[0.0, 1.0]$, perform random perturbation in the normalized range, and renormalize the resulting value back to the original range.

We implemented FIRE-PBT from scratch, as no implementation was available. PBT was also implemented from scratch. As to PB2, PB2-Mix, and BG-PBT, we built upon the official implementations and adapted implementations of PB2-Mix and BG-PBT to be task-agnostic. Furthermore, we have fixed several issues with the gradients of PB2 and PB2-Mix that were present in these implementations (the issues were confirmed by the original authors in private communication). Specifically:

- PB2: The gradient of variance should not square the length scale, as it is not squared in the forward path of the kernel (`https://github.com/jparkerholder/PB2/blob/master/kernel.py`, lines 31, 47). Alternatively, the length scale could be squared in both places.

- PB2: The gradient of variance should be multiplied by the value of the time kernel, which is independent of it and is multiplied by it in the forward path (`https://github.com/jparkerholder/PB2/blob/master/kernel.py`, lines 33, 47).

- PB2: The gradient of the length scale has an erroneous additional factor of "-2", which can be confirmed by taking the gradient of the kernel with respect to the length scale (`https://github.com/jparkerholder/PB2/blob/master/kernel.py`, line 48).

- PB2-Mix: The gradient of the length scale has an erroneous minus sign, which is absent in the gradient formula in the PB2-Mix publication itself (`https://github.com/jparkerholder/procgen_autorl/blob/main/pb2_utils.py`, line 248).

Based on our preliminary experiments, these changes did not impact the final results achieved by the algorithms but substantially improved their exploration speed. This happened because the errors led to some gradients having signs opposite to the correct ones, which deteriorated the performance of L-BFGS that relied on these analytical gradients to optimize the hyperparameters of the kernel.

## E   Search spaces

In classification tasks, the *small* search space corresponds to searching just the learning rate. In the *medium* space, the number of augmentations in RandAugment and the augmentation strength are additionally considered. The *large* space also includes weight decay and momentum. See Table 1.

In RL tasks, our search space is based on that of Wan et al. (2022), with architecture-related hyperparameters omitted. We also narrow the ranges of the "batch size" and "number updates per epoch" hyperparameters in

| Hyperparameter | Type | Exponent base | Range |
|---|---|---|---|
| learning rate | real | 10 | [-6, 0] |
| number of augmentations | integer | ✗ | [1, 4] |
| augmentation strength | integer | ✗ | [1, 30] |
| weight decay | real | 10 | [-8, -2] |
| momentum | real | ✗ | [0.5, 0.999] |

Table 1: Hyperparameter tuning ranges for the classification tasks. All variables above the first and the second dashed lines are included in the *small* and *medium* search spaces accordingly, while the full table corresponds to the *large* search space.

order to reduce the computational burden of the experiments. We refer to the resulting search space as *large*. The *medium* space is its subset that contains only "learning rate", "entropy coefficient", and "batch size". The *small* space contains "learning rate" only. See Table 2.

| Hyperparameter | Type | Exponent base | Range |
|---|---|---|---|
| learning rate | real | 10 | [-4, -3] |
| entropy coefficient | real | 10 | [-6, -1] |
| batch size | integer | 2 | [8, 10] |
| discount factor | real | ✗ | [0.9, 0.9999] |
| unroll length | integer | ✗ | [5, 15] |
| reward scaling | real | ✗ | [0.05, 20] |
| number of updates per epoch | integer | ✗ | [2, 16] |
| GAE parameter | real | ✗ | [0.9, 1] |
| clipping parameter | real | ✗ | [0.1, 0.4] |

Table 2: Hyperparameter tuning ranges for the RL tasks. All variables above the first and the second dashed lines are included in the *small* and *medium* search spaces accordingly, while the full table corresponds to the *large* search space.

## F  Statistical analysis

Due to high computational costs and the large number of settings considered in this article, we were able to run each algorithm with only a few random seeds in each setting: 5 seeds for the classification tasks, 7 seeds for the RL tasks. To partially counteract the lack of data, the results of the classification tasks were pulled together (so that each of the algorithm had 10 data points), and the results of the RL tasks were pulled together (so that each algorithm had 14 data points).

The tests were performed independently in each setting, with the target p-value of 0.05. Firstly, a Friedman test (Friedman, 1937) was performed to determine if any algorithm achieved results statistically different to those of the others. If so, pairwise signed Wilcoxon rank tests (Wilcoxon, 1992) with Holm correction (Holm, 1979) were performed. The results are presented in Figure 10 and referenced in the main text. We would like to additionally speculate that repeating experiments more times could lead to more significant differences found in some cases with large differences in average ranks (e.g., between PB2 and FIRE-PBT in Figure 10 (a) with 3 steps).

## G  Additional results on Tiny ImageNet

In Section 5.8, we unexpectedly found that FIRE-PBT does not outperform other PBTs on Tiny ImageNet in the setting where it did so for all other datasets (Fashion-MNIST, CIFAR-10/100): *large* search space, 100 outer `steps`. We hypothesized that this could be due to the increased difficulty of the underlying task

(a) Classification tasks: vary number of steps

(b) RL tasks: vary number of steps

(c) Classification tasks: vary search space

(d) RL tasks: vary search space

(e) Classification tasks: vary population size

(f) RL tasks: vary population size

Figure 10: Critical difference diagrams for PBTs. Algorithms within a bracket were not found to be statistically significantly different from each other. Each row in each subfigure corresponds to one experimental setting that was analyzed independently of all others.

in combination with the *large* search space. To test the impact of the search space, we ran an additional experiment in the *small* search space. Figure 11 shows that in this setting FIRE-PBT outperforms other variants, as expected.

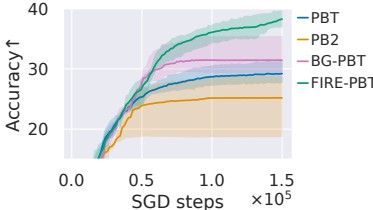

Figure 11: The performance (IQM, with IQR shaded) of PBTs on Tiny ImageNet in the *small* search space.

## H   Results on the test split

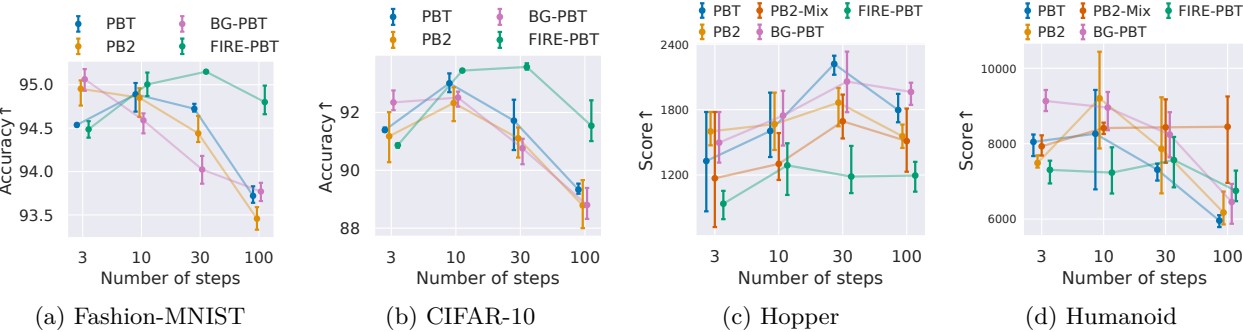

Figure 12: Test performance (IQM and IQR) of the PBT variants as the number of outer `steps` is varied, while the number of inner steps (SGD steps for classification, environment steps for RL) is kept constant.

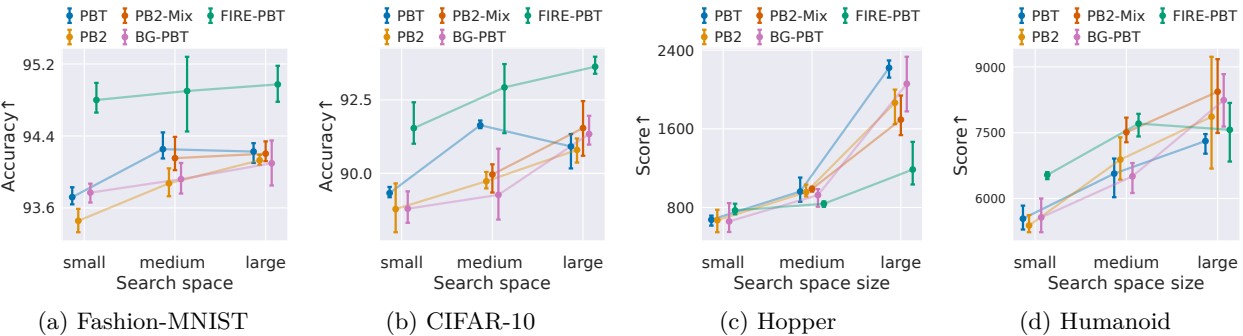

Figure 13: Test performance (IQM and IQR) of the PBT variants as the search space size is increased. Some IQR marks are invisible due to IQRs being too narrow.

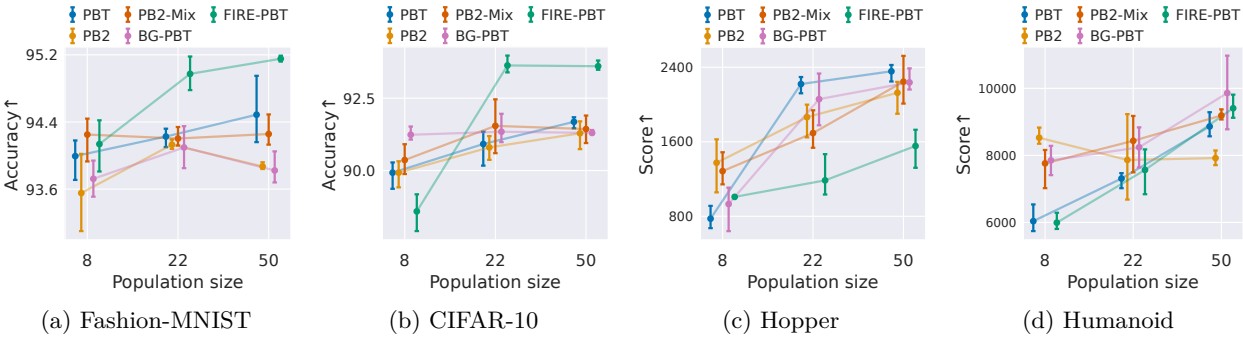

Figure 14: Test performance (IQM and IQR) of the PBT variants as the population size is increased.

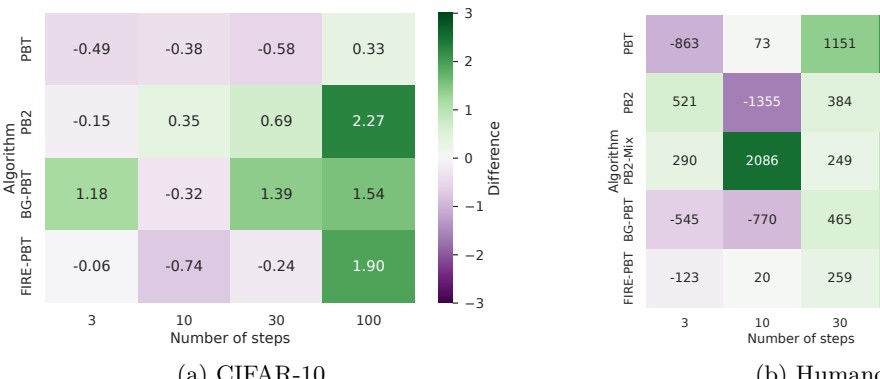

(a) CIFAR-10                (b) Humanoid

Figure 15: Differences in IQMs of the performance of the PBT variants when changing the value of $\lambda$ of truncation selection from 25% to 12.5%.

## I  Impact of $\lambda$ on performance of the PBT variants

To evaluate the impact of the truncation selection parameter $\lambda$ on the performance of PBTs, we repeat a subset of experiments from Figure 5 (specifically, those with CIFAR-10 and Humanoid) with the value of $\lambda$ of truncation selection changed from the default 25% to 12.5%.

Figure 15 shows the difference in IQM that arise from such a change of $\lambda$. It can be generally observed that decreasing lambda improves results for some combinations of (algorithm, number of steps) but leads to worse results in others. Therefore, we cannot conclude that one of the considered values of lambda should be generally preferred.

Interestingly, all algorithms seem to benefit from the decreased lambda when the number of steps is 100. Since smaller $\lambda$ corresponds to lower selection pressure and slower evolution, reducing it could be especially helpful in reducing short-term focus of optimization in this setting of many steps. We leave further investigation of the interaction between the hyperparameters and the performance of PBTs for future research.

## J  Impact of the perturbation factor on performance of FIRE-PBT in challenging search spaces

In order to evaluate the potential impact of the perturbation factor hyperparameter of FIRE-PBT on its performance in larger and more challenging search spaces, we have run FIRE-PBT with the perturbation factor of 1.25 instead of 2.0 in the *large* search space on two RL tasks, Hopper and Humanoid. As can be seen in Figure 16, the smaller perturbation factor leads to a decreased score on Hopper and an increased score on Humanoid. This indicates that the perturbation-based search procedure is sensitive to the perturbation factor, which may need to be tuned per task to achieve optimal performance.

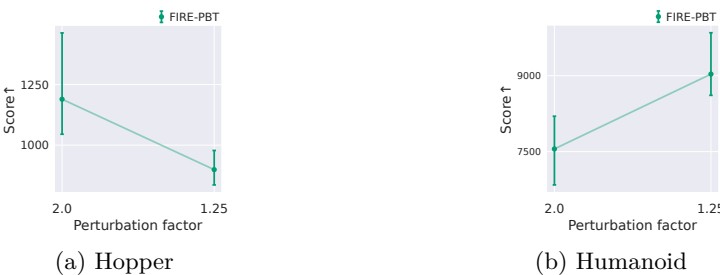

(a) Hopper                (b) Humanoid

Figure 16: The performance (IQM and IQR) of FIRE-PBT when the perturbation factor is reduced from 2.0 to 1.25. The *large* search space is used.

