# OpenReview forum: "To Be Greedy, or Not to Be – That Is the Question for Population Based Training Variants"
_TMLR — Accepted by TMLR_

### Review · Reviewer_rR61 · 2025-02-28

**Summary Of Contributions:**

The authors empirically evaluate five recent Population Based Training (PBT) algorithms on classification and reinforcement learning tasks to provide a fair comparison of PBT algorithms on a unified set of tasks. It is shown that, although there is no clear winner, perturbation-based PBT variants perform better on classification tasks, while PBT algorithms leveraging Bayesian Optimization for candidate selection perform better on RL tasks. Besides the empirical results, the authors provide theoretical insights that partially explain their observations.

**Audience:**

Yes

**Broader Impact Concerns:**

n.a.

**Claims And Evidence:**

Yes

**Requested Changes:**

**Shorten Sec. 4.1 and fix minor flaws**

- after Eq. 6, the authors write: `Observe that the argument of argmax must be the entire schedule for the transitions in the original proof (Eq. 4) to hold as written. Thus, the lemma is proved in the non-sequential setting.` I agree that the Lemma is shown for the non-sequential setting. However, this is not because the entire schedule is the argument of argmax. Rather, it is due to the dependencies among the terms in the summation (as correctly described below). This should be changed.
- Sec. 4.1 can be significantly shortened. In my opinion, the authors could discuss the dependencies among the terms in the sum of Eq. 6 briefly below the equation, saying that greedy optimization does not guarantee to find the maximum of Eq. 6. After that, the authors could briefly state that PBT is not a purely greedy optimization algorithm (3-5 lines should be enough on that). This would increase clarity and readability.

**Other Changes**

- Sec. 4.4 should be moved to Sec. 5. Also, the authors should discuss the design of the toy functions a bit more and provide justification for their design. Additionally, I believe a slightly more complex PlainToy problem is required to validate the theoretical findings (as described in "Weaknesses and Questions").
- if computationally feasible, a senstivity analysis of the hyperparameters of (at least some) PBT algorithms would be beneficial to see the influence of these parameters on the overall performance.

**Strengths And Weaknesses:**

**Strengths**

- the authors define a common set of tasks to evaluate PBT algorithms fairly, thereby defining a standard experimental setup that other works can adopt in the future
- an extensive empirical evaluation is conducted, showing that although recent works on PBT claim to achieve new state-of-the-art results, no clear winner can be identified among those methods
- a theoretical analysis is provided, showing that the optimization dynamics of PBT algorithms have to be properly accounted for in future theoretical analysis
- overall, the paper is well-structured and written clearly

**Weaknesses and Questions**

- Sec. 4.1 can be shortened significantly to improve readability (see Requested Changes)
- the work does not propose an extended theoretical framework that can be used to analyze the optimization properties of PBT algorithms
- the PlainToy problem introduced to support the theoretical findings is too simplistic, in my opinion: It is quadratic in $\theta$ (thus convex) and linear in $h$. Another more complex problem would significantly strengthen the work.
- Fig. 3(a): it is unclear to me, why PB2 is so much worse compared to PBT although both show similar behavior in decreasing the learning rate
- Fig. 4: Why is the standard deviation of BO-based PBT algorithms increasing towards the end (see BG-PBT in (a) and PB2 in (b))?
- Fig. 6(c, d): It seems that the results in Fig. 6(c) and (d) contradict the results in Fig. 4: In Fig. 4, BG-PBT was best on Hopper and PB2 was best on Humanoid. However, in Fig. 6, none of them is best for any search space size. Could the authors explain this?
- I would have expected FIRE-PBT to shine on RL tasks as it focuses on optimization on long-term gains. However, the greedier BO-PBT variants seem to perform better on RL tasks. Do the authors have an explanation for this?

---

> ### Author Response · Authors · 2025-04-03
> **Response to reviewer rR61 [Part 1]**
>
> > [Weaknesses] Sec. 4.1 can be shortened significantly to improve readability (see Requested Changes)
>
> > [Requested changes] Shorten Sec. 4.1 and fix minor flaws
>
> We have modified Sec. 4.1 based on the reviewer's feedback, see below for details.
>
> > [Requested changes] after Eq. 6, the authors write: `Observe that the argument of argmax must be the entire schedule for the transitions in the original proof (Eq. 4) to hold as written. Thus, the lemma is proved in the non-sequential setting.` I agree that the Lemma is shown for the non-sequential setting. However, this is not because the entire schedule is the argument of argmax. Rather, it is due to the dependencies among the terms in the summation (as correctly described below). This should be changed.
>
> We agree that the fundamental cause of the issue is the dependencies between the summands. We further believe that these dependencies require the entire schedule to be the argument of the argmax (without the dependencies, the entire schedule wouldn't have to be the argument). To address this in the article, we have added a clarification to that paragraph, which now reads: "Observe that **due to the dependencies between the summands**, the argument of argmax must be the entire schedule for the transitions in the original proof (Eq. 3) to hold as written."
>
> > [Requested changes] In my opinion, the authors could discuss the dependencies among the terms in the sum of Eq. 6 briefly below the equation, saying that greedy optimization does not guarantee to find the maximum of Eq. 6. After that, the authors could briefly state that PBT is not a purely greedy optimization algorithm (3-5 lines should be enough on that). This would increase clarity and readability.
>
> To increase clarity and readability, we have condensed the second half of the section that concerns greediness into one paragraph. We would like to keep the first half largely as is, since we see value in explicitly laying out problems that were tricky enough to go unnoticed by the original authors and peer reviewers. We hope that this could be useful for future researchers of this topic.
>
> > [Weaknesses] the PlainToy problem introduced to support the theoretical findings is too simplistic, in my opinion: It is quadratic in $\theta$ (thus convex) and linear in $h$. Another more complex problem would significantly strengthen the work.
>
> > [Requested changes] Sec. 4.4 should be moved to Sec. 5. Also, the authors should discuss the design of the toy functions a bit more and provide justification for their design. Additionally, I believe a slightly more complex PlainToy problem is required to validate the theoretical findings (as described in "Weaknesses and Questions").
>
> We agree that Sec. 4.4 (describing the toy problems) fits better in Sec. 5, we have moved it.
>
> Next, we would like to clarify the idea behind PlainToy, which is to show that if the problem is such that greedy behavior is optimal, then the greedier & more effective BO PBTs will reach better results than the less greedy PBTs. In our opinion, PlainToy achieves this design goal and serves as straightforward contrast to TimeLinkedToy, where greedy behavior is detrimental and leads to BO PBTs performing worse. In the spirit of designing understandable toy problems, we would like to keep the PlainToy problem as simple as possible to avoid any complexity that is not relevant to the purpose of the problem and that could make the analysis less clear. We have added the motivation behind PlainToy to the Section 5.2.
>
> > [Weaknesses] Fig. 3(a): it is unclear to me, why PB2 is so much worse compared to PBT although both show similar behavior in decreasing the learning rate
>
> While we agree that PBT and PB2 show somewhat similar behavior in decreasing the learning rate (LR), it can be seen in Fig. 3(a) (right) that in the initial phase of the training (approx. between steps 5,000 - 12,000), PB2 drops the LR much faster (keeping in mind that the y-axis is in log scale). Consistently with this, PB2 performs visibly better than PBT during this period but suffers from this greedy choice in the later stages of the training.
>
> [due to character limit, the response is continued in Part 2]

---

> ### Author Response · Authors · 2025-04-03
> **Response to reviewer rR61 [Part 2]**
>
> > [Weaknesses] Fig. 4: Why is the standard deviation of BO-based PBT algorithms increasing towards the end (see BG-PBT in (a) and PB2 in (b))?
>
> The interquartile range can be narrower at the start - when runs with all seeds improve at about the same speed - than towards the end, when some runs stagnate while others continue improving. We have checked that this is what happens with Bayesian PBTs in Fig. 4. Note that this phenomenon is not restricted to Bayesian PBTs: for example, FIRE-PBT in Fig. 4 (b) (Humanoid) also exhibits it (to a degree).
>
> > [Weaknesses] Fig. 6(c, d): It seems that the results in Fig. 6(c) and (d) contradict the results in Fig. 4: In Fig. 4, BG-PBT was best on Hopper and PB2 was best on Humanoid. However, in Fig. 6, none of them is best for any search space size. Could the authors explain this?
>
> The reviewer is correct that the results in Fig. 4 and Fig. 6 are inconsistent. This is due to the results in Fig. 4 coming from an experiment with 100 outer steps, while the results in Fig. 6 are from an experiment with 30 outer steps. We have added the corresponding plots with 30 outer steps to Fig. 4, they are consistent with Fig. 6.
>
> > [Weaknesses] I would have expected FIRE-PBT to shine on RL tasks as it focuses on optimization on long-term gains. However, the greedier BO-PBT variants seem to perform better on RL tasks. Do the authors have an explanation for this?
>
> While the precise reason is difficult to establish (given how many random factors influence both RL and the PBTs), we would like to hypothesize that FIRE-PBT suffers from its relatively poor perturbation-based exploration that could be insufficient for larger search spaces of the RL tasks (that are also less well-behaved than the classification tasks). Consider that in the original FIRE-PBT paper (Dalibard & Jaderberg, 2021), only one or two hyperparameters are optimized, whereas in most of our RL experiments, the large search space with 9 hyperparameters is used. It can also be seen in Fig. 6 (c, d) that on RL tasks with the small search space of a single hyperparameter, FIRE-PBT achieves (marginally) better IQM than all other algorithms.
>
> > [Requested changes] If computationally feasible, a senstivity analysis of the hyperparameters of (at least some) PBT algorithms would be beneficial to see the influence of these parameters on the overall performance.
>
> Done. We perform an additional small-scale sensitivity analysis by considering a different value of $\lambda$ of truncation selection. It is the only hyperparameter that is shared by all PBTs and that has not been previously explored in our paper (the other shared hyperparameters are the population size and the number of outer steps). We repeat a subset of experiments in Fig. 5 (where the step size is varied) with $\lambda$=12.5% rather than 25% and report the results in Appendix I.
>
> To summarize, we find that decreasing $\lambda$ improves results for some combinations of (algorithm, number of steps) but not others. Interestingly, it seems to be beneficial for all algorithms when the number of steps is 100. Since smaller $\lambda$ corresponds to lower selection pressure and slower evolution, reducing it could be especially helpful in reducing short-term focus in this setting of many steps.
>
> We have also extended Section 6 with a statement that "deeper investigation of the interaction between the hyperparameters, the performance, and the behavior of PBTs is a potentially interesting subject for future work."
>
> References:
>
> (Dalibard & Jaderberg, 2021) Valentin Dalibard and Max Jaderberg. Faster Improvement Rate Population Based Training, September 2021. URL http://arxiv.org/abs/2109.13800. arXiv:2109.13800

---

> > ### Comment · Reviewer_rR61 · 2025-04-09
> >
> > I thank the authors for the thorough rebuttal.
> >
> > Most of my concerns have been resolved.
> >
> > I only have one remark left:
> >
> > > FIRE-PBT suffers from its relatively poor perturbation-based exploration that could be insufficient for larger search spaces of the RL tasks [...]
> >
> > To support this hypothesis, it would be beneficial to have experiments on one or two tasks showing that the default mutation rate of FIRE-PBT is not optimal for larger search spaces.

---

> > > ### Author Response · Authors · 2025-04-11
> > > **Response to reviewer rR61**
> > >
> > > We are glad to hear that most of the reviewer's concerns have been resolved.
> > >
> > > In order to evaluate the potential impact of the perturbation factor hyperparameter of FIRE-PBT on its performance in larger and more challenging search spaces, we have run FIRE-PBT with the perturbation factor of 1.25 instead of 2.0 in the large search space on two RL tasks, Hopper and Humanoid. As can be seen in Appendix J of the newly updated manuscript, the smaller perturbation factor leads to a decreased score on Hopper and an increased score on Humanoid. This indicates that the perturbation-based search procedure is sensitive to the perturbation factor, which may need to be tuned per task to achieve optimal performance.

---

> > > > ### Comment · Reviewer_rR61 · 2025-04-14
> > > >
> > > > Thank you for pointing me to App. J. My concerns are resolved.

---

### Review · Reviewer_Tn4y · 2025-03-19

**Summary Of Contributions:**

The authors compared the performance of some variants of PBTs on a set of image classification and reinforcement learning tasks for hyperparameter autotuning. Theoretical analysis and discussion are introduced to explain the observed differences in the performance of different approaches.

**Audience:**

Yes

**Broader Impact Concerns:**

No.

**Claims And Evidence:**

Yes

**Requested Changes:**

* In the paragraph after (6), the authors mentioned that the transition (6.3) holds for any schedule that $f_t(h_t^A)$ is constant for all $t$. Maybe the authors could include an specific example for such a schedule? It would me much helpful for the readers to understand.
* In the last paragraph of section 4.1, the authors mentioned that the schedules discoverd by Bayesian PBTs  asymptotically approach {$\tilde{h}_1, \ldots, \tilde{h}_T$}. This is not quite clear to me according to the analysis in the text. Is this discussed in the paper from Parker-Holder et al., or did I missed something? It would be good if the authors could explain this point in more detail.
* The authors discussed the limitations of the current theoretical approach in section 4.2, but it is not quite clear to me what is the author's motivation of writing this section. Is it just for some discussion, or related to the subsequent experiments? Could the authors make their claim in this section more clear?
* Not all equations in display mode are properly punctuated. As far as I noticed, except for (3), a period is missing for all other display mode equations.

Minor:
* It is a bit weird for me to put the toy examples in the analysis section. The authors could restruct the sections about this part such that the structure of the whole paper is more clear.
* All equations in this paper are numbered but not all of them are referred to somewhere else. Particularly, the equations (5) and (6) should be understood as the same equation but are numbered twice. The authors could consider only numbering those equations that are used, and remove duplicate numbering.

**Strengths And Weaknesses:**

*Strengths*

This paper evaluates the performance of different PBT variants for hyperparameter autotuning in multiple tasks. The authors made multiple claims about the properties of different methods and the experiments are well designed to support their claims. The authors also tried to interprete their observations from a theoretical perspective.

*Weaknesses*

* I have some questions about the analysis section, see the requested changes below.
* I should mention that I am not an expert in AutoML. According to my background knowledge (I am from convex optimization and inverse RL), I can not really evaluate wheather the PBT variants compared in this paper are representative or thorough. I think this should be evaluated by the other reviewers.

---

> ### Author Response · Authors · 2025-04-03
> **Response to reviewer Tn4y**
>
> > In the paragraph after (6), the authors mentioned that the transition (6.3) holds for any schedule $f_t(h^A_t)$ that is constant for all $t$. Maybe the authors could include an specific example for such a schedule? It would me much helpful for the readers to understand.
>
> Sorry for the confusion, we did not mean to imply that the schedule or the $f_t$ is constant, we would like to clarify that paragraph. We write that transition (6.3) holds because $f_t(\tilde{h}_t)$ is constant. We mean that $f_t(\tilde{h}_t)$ is a specific real value at any time $t$, not that this value is the same for all times t = 1, ..., T.
>
> The value at any time $t$ is a constant because $\tilde{h}_t$ is a specific schedule that is (greedily) optimal for the problem (by definition of $\tilde{h}_t$). However, we note in our paper that the transition (6.3) will also hold not just for $\tilde{h}_t$ but for any fixed schedule, because for any fixed schedule, a value at time t will be just a constant value that is independent of $h_t$ (without tilde) that are the arguments of argmax.
>
> To clarify our meaning, we have reworded a part of the relevant paragraph from "$\forall t\ f_t(\tilde{h}_t)$ is constant" to "for each time $t$, $f_t(\tilde{h}_t)$ is a constant", we hope that this is helpful. We believe adding an example would not be helpful, since any arbitrary schedule can be used as $h^A_t$.
>
> > In the last paragraph of section 4.1, the authors mentioned that the schedules discoverd by Bayesian PBTs asymptotically approach $\{\tilde{h}_1, \ldots, \tilde{h}_T\}$. This is not quite clear to me according to the analysis in the text. Is this discussed in the paper from Parker-Holder et al., or did I missed something? It would be good if the authors could explain this point in more detail.
>
> The reviewer is right that this point is not addressed in our paper and that it is discussed in (Parker-Holder et al, 2020). While reexamining (Parker-Holder et al, 2020) to provide a specific reference to the reviewer, we have noticed that the relevant formulation is not entirely consistent with ours. Specifically, per (Parker-Holder et al, 2020), "the gap between $f_t(x_t)$ and the optimal $f_t(x^*_t$ ) vanishes asymptotically using PB2" (subject to conditions), which is different from our claim that the schedule $h_t$ approaches $\tilde{h}_t$ (where $x_t$ is $h_t$ in our notation and $x^*_t$ is $\tilde{h}_t$). This is an oversight on our part. We adjust our wording from "the schedules approach $\tilde{h}_t$" to "the returns of the schedules approach those of $\tilde{h}_t$" and add a citation of (Parker-Holder et al, 2020).
>
> > The authors discussed the limitations of the current theoretical approach in section 4.2, but it is not quite clear to me what is the author's motivation of writing this section. Is it just for some discussion, or related to the subsequent experiments? Could the authors make their claim in this section more clear?
>
> We find it important to highlight how the theoretical results (both ours and of previous work) are expected to differ from practical results, due to some inherent limitations. A reader guided by purely theoretical results could be surprised by the empirical ones. Therefore, in section 4.2, we mention the limitations of the theoretical results: (1) the implications of the population-based optimization are not fully considered, (2) the formalization of the problem is not aligned with the declared goal of optimizing the final performance (i.e., the issues with the theoretical approach run deeper than the problems with Lemma 1 considered in the previous subsection), (3) how the formalization of the problem is connected to Perturbation PBTs.
>
> > Not all equations in display mode are properly punctuated. As far as I noticed, except for (3), a period is missing for all other display mode equations.
>
> Done. We have added missing periods to all display mode equations.
>
> > It is a bit weird for me to put the toy examples in the analysis section. The authors could restruct the sections about this part such that the structure of the whole paper is more clear.
>
> Done. We have moved the toy problem subsection to section 5, as it is indeed more related to experiments than analysis.
>
> > All equations in this paper are numbered but not all of them are referred to somewhere else. Particularly, the equations (5) and (6) should be understood as the same equation but are numbered twice. The authors could consider only numbering those equations that are used, and remove duplicate numbering.
>
> Done. We have renumbered all equations, removing the numbers of unused equations and having a single number (4) corresponding to the equation that was previously referred to by both (5) and (6).
>
> References:
>
> (Parker-Holder et al, 2020) Jack Parker-Holder, Vu Nguyen, and Stephen J Roberts. Provably Efficient Online Hyperparameter Optimization with Population-Based Bandits. In Advances in Neural Information Processing Systems, volume 33, pp. 17200–17211.

---

### Review · Reviewer_2tGy · 2025-03-30

**Summary Of Contributions:**

The paper studies Population based training (PBT) methods which is one class of algorithms for dynamic hyper-parameter optimization (HPO) in neural networks. First, the paper studies and analyzes the assumptions behind the theoretical analysis of a particular algorithm in this class (PB2 Parker-holder et al). Second, the paper empirically analyzes the performance of multiple algorithm variants in the broader PBT class on a series of image classification and reinforcement learning HPO setting.

**Audience:**

Yes

**Broader Impact Concerns:**

I do not see any ethical concern.

**Claims And Evidence:**

No

**Requested Changes:**

Please see strengths and weaknesses section.

**Strengths And Weaknesses:**

- The analysis clarifying the assumptions of cumulative regret analysis from PB2 is a good sound contribution and put forths a hypothesis that Bayesian PBTs work by optimizing greedily.

- The related work coverage in the paper is excellent which makes it clear to understand the landscape of PBT methods.

- I like the extended discussion about step size which is typically missing in existing work and one big source of inconsistencies in comparisons of different PBT algorithms.

- The paper clarifies that other Bayesian PBTs (like BG-PBT Wan et al) are out of scope of the paper but it does seem to weaken the theoretical analysis in 4.1 since it focuses on one very specific paper within this class.

- I am most concerned about the experimental analysis in the paper which is inconsistent and doesn't seem to follow nicely with the hypotheses made in the theoretical analysis section. For example, in Figure 4, Bayesian PBTs perform better than perturbation ones for reinforcement learning tasks which is attributed to greedy optimization working better for these tasks. This is counterintuitive since RL tasks have a larger search space than image classification one and one would expect to larger spaces to be less amenable to greedy optimization.

- The results in section 5.7 show that there is no clear method that performs clearly better. It is not clear how can we leverage the greedy optimization hypothesis to explain the performance difference here. Again, it is slightly unclear whether the theoretical analysis helps in understanding the results.

- In the light of the results, I think the paper should consider either studying or adding a recommendation for future work regarding studying other class of algorithms (for example, RL based or hypergradient methods) for dynamic HPO going beyond PBT. In fact, based on this paper, all issues mentioned with these approaches in the related work section (like meta-hyperparameters of their own or inefficiency) seem to arise with PBT methods as well.

---

> ### Author Response · Authors · 2025-04-03
> **Response to reviewer 2tGy [Part 1]**
>
> > The paper clarifies that other Bayesian PBTs (like BG-PBT Wan et al) are out of scope of the paper but it does seem to weaken the theoretical analysis in 4.1 since it focuses on one very specific paper within this class.
>
> We would like to clarify that the original theoretical analyses of both PB2-Mix and BG-PBT build upon the analysis of PB2, and therefore our analysis applies to these algorithms as well. When we write in Sec. 3.2 that "BG-PBT places focus on enabling neural architecture search, which is out of scope for our article, we therefore omit the related adaptations from our analysis and experiments", we mean specifically the omission of the architecture search part (which is also omitted from the analysis of BG-PBT). Our analysis remains applicable to a variant of BG-PBT without architecture search, analysis of which in the original BG-PBT paper (Wan et al., 2022) builds upon the analysis of PB2. Therefore, we disagree that our results are limited to a single algorithm.
>
> To make the paper clearer, we have extended the section describing BG-PBT with a statement that our analysis and experiments apply to a variant of BG-PBT without architecture search. We have also modified the last paragraph of Section 4.1 to read "In the theory **underlying all** Bayesian PBTs" in place of "In the theory of Bayesian PBTs".
>
> > I am most concerned about the experimental analysis in the paper which is inconsistent and doesn't seem to follow nicely with the hypotheses made in the theoretical analysis section. For example, in Figure 4, Bayesian PBTs perform better than perturbation ones for reinforcement learning tasks which is attributed to greedy optimization working better for these tasks. This is counterintuitive since RL tasks have a larger search space than image classification one and one would expect to larger spaces to be less amenable to greedy optimization.
>
> We agree that our theoretical contributions do not fully explain the empirical results. As mentioned in Section 4.2, both previous theory and our correction of it do not fully correspond to what PBTs do in practice (e.g., the selection procedure is ignored despite its potential importance). Due to these limitations, we can only expect for the theory to shed some light on the empirical behavior, not fully explain it.
>
> We confirmed that our analysis has some explanatory value: on toy problems with understandable dynamics, the algorithms behave in accordance with the analysis. Furthermore, on image classification problems where only the Learning Rate (LR) is optimized, we also generally see behavior consistent with the analysis. Unfortunately, the results are indeed less clear for larger search spaces and for RL. However, this should not be too unexpected, considering how much complexity and randomness becomes involved, especially in RL (the environment dynamics, the PPO agent, and a PBT variant on top).
>
> We would also like to note that while we refer to Bayesian PBTs as "greedier" (and Perturbation PBTs as "less greedy"), none of them are just greedy algorithms. Asymptotically, Bayesian PBTs are guaranteed to approach the returns of a greedy schedule, but in practice, Bayesian PBTs operate in finite time frames and with finite population sizes. It is not very surprising to us that their more effective Bayesian optimization (that was originally designed and evaluated mostly on RL tasks) can achieve better results on RL tasks. Their greedy behavior when tuning LR of an image classifier **was** surprising, we hope that our analysis makes this less surprising to the readers.
>
> The Perturbation PBTs, while theoretically less greedy than Bayesian PBTs, can still be more or less greedy (as shown by our experiments where the step size is varied). Additionally, their less effective exploration procedure is not guaranteed to reduce greediness, it can simply be a worse optimization procedure that struggles to perform in more complicated search spaces and tasks. Less greedy & less effective algorithm may underperform the greedier but more effective algorithm even if greediness is detrimental (and so a hypothetical "best" algorithm would be non-greedy & effective).
>
> To sum up, we believe that our theory has some explanatory value and can be useful in some circumstances, but is also subject to limitations, which we fully acknowledge in our conclusions in Sec. 6. The tasks being optimized add even more complexity and uncertainty to the behavior of the algorithms. Better understanding of the tasks, the algorithms, and their interaction is clearly a valuable direction for future research.
>
> [due to character limit, the response is continued in Part 2]

---

> ### Author Response · Authors · 2025-04-03
> **Response to reviewer 2tGy [Part 2]**
>
> > The results in section 5.7 show that there is no clear method that performs clearly better. It is not clear how can we leverage the greedy optimization hypothesis to explain the performance difference here. Again, it is slightly unclear whether the theoretical analysis helps in understanding the results.
>
> We agree that based on our experimental results, there appears to be no single dominant PBT variant, which is one of the main conclusions of our paper. As mentioned in Part 1 of the response, we believe that our theoretical analysis can provide only a partial explanation that may have more or less explanatory power in various settings (e.g., it seems to apply well to tuning only the LR of an image classifier). Nonetheless, we believe that having a limited explanation can still be valuable and can serve as a foundation of a more full understanding in the future. Moreover, we consider our analysis useful since it points out (what we consider to be) problems with previously published theoretical results.
>
> > In the light of the results, I think the paper should consider either studying or adding a recommendation for future work regarding studying other class of algorithms (for example, RL based or hypergradient methods) for dynamic HPO going beyond PBT. In fact, based on this paper, all issues mentioned with these approaches in the related work section (like meta-hyperparameters of their own or inefficiency) seem to arise with PBT methods as well.
>
> We agree that other classes of algorithms deserve to be studied in the context of dynamic HPO. Since we intentionally focus our paper on analysis and evaluation of PBT variants, we consider comparisons with other algorithms as out-of-scope.
>
> We have extended our conclusion that PBTs can suffer from inefficiency with the following recommendation: "In order for a more complete picture of the upsides and downsides of different approaches to dynamic HPO to emerge, alternative dynamic HPO algorithms should be studied and compared both to PBTs and to each other."
>
> References:
>
> Xingchen Wan, Cong Lu, Jack Parker-Holder, Philip J. Ball, Vu Nguyen, Binxin Ru, and Michael Osborne. Bayesian Generational Population-Based Training. In Proceedings of the First International Conference on Automated Machine Learning, pp. 14/1–27. PMLR, September 2022. URL https://proceedings.mlr.press/v188/wan22a.html. ISSN: 2640-3498.

---

### Author Response · Authors · 2025-04-03
**Response to all reviewers**

We thank all the reviewers for their valuable comments that helped us improve the paper. We have uploaded the updated paper, with the changes highlighted. Below, we individually respond to the specific points raised by each reviewer.

---

### Decision · Action_Editor_B7Jr · 2025-05-19

**Recommendation:** Accept as is

**Comment:**

The paper provides both a theoretical and empirical analysis of population-based training (PBT) to dynamically optimize the hyperparameters of supervised training for deep neural networks and deep reinforcement learning agents. It compares multiple methods from the literature, showing that none of the methods performs consistently better across all benchmarks.
Results indicate that Bayesian optimization-based methods behave more greedily than random search-based counterparts, such as vanilla PBT, which performs better on certain benchmarks that favor short-term gains.

The reviewers unanimously recommend acceptance of the paper, appreciating its insights into the behavior of the different methods. They did, however, point out some discrepancies between the theoretical and empirical results. These discrepancies are explicitly acknowledged and discussed in the paper. While the paper does not fully explain all empirical findings in detail, it nonetheless offers valuable insights for the TMLR community.

**Audience:**

Hyperparameter optimization for reinforcement learning and supervised machine learning is a vivid field and hence I confident that individuals in TMLR's audience appreciate the paper.

**Claims And Evidence:**

The paper provides sufficient empirical and theoretical evidence for its claims.